# ZiCo: Zero-shot NAS via inverse Coefficient of Variation on Gradients

**Guihong Li[1], Yuedong Yang[1], Kartikeya Bhardwaj[2],\*, Radu Marculescu[1]**

[1]The University of Texas at Austin, [2]Qualcomm
{lgh,albertyoung,radum}@utexas.edu, kbhardwa@qti.qualcomm.com

## ABSTRACT

Neural Architecture Search (NAS) is widely used to automatically obtain the neural network with the best performance among a large number of candidate architectures. To reduce the search time, zero-shot NAS aims at designing *training-free* proxies that can predict the test performance of a given architecture. However, as shown recently, none of the zero-shot proxies proposed to date can actually work consistently better than a naive proxy, namely, the number of network parameters (#Params). To improve this state of affairs, as the main theoretical contribution, we first reveal how some specific gradient properties across different samples impact the convergence rate and generalization capacity of neural networks. Based on this theoretical analysis, we propose a new zero-shot proxy, *ZiCo*, the first proxy that works consistently better than #Params. We demonstrate that ZiCo works better than State-Of-The-Art (SOTA) proxies on several popular NAS-Benchmarks (NASBench101, NATSBench-SSS/TSS, TransNASBench-101) for multiple applications (e.g., image classification/reconstruction and pixel-level prediction). Finally, we demonstrate that the optimal architectures found via ZiCo are as competitive as the ones found by one-shot and multi-shot NAS methods, but with much less search time. For example, ZiCo-based NAS can find optimal architectures with 78.1%, 79.4%, and 80.4% test accuracy under inference budgets of 450M, 600M, and 1000M FLOPs, respectively, on ImageNet within 0.4 GPU days. Our code is available at https://github.com/SLDGroup/ZiCo.

## 1 INTRODUCTION

During the last decade, deep learning has achieved great success in many areas, such as computer vision and natural language modeling Krizhevsky et al. (2012); Liu & Deng (2015); Huang et al. (2017); He et al. (2016); Dosovitskiy et al. (2021); Brown et al. (2020); Vaswani et al. (2017). In recent years, neural architecture search (NAS) has been proposed to search for optimal architectures, while reducing the trial-and-error (manual) network design efforts Baker et al. (2017); Zoph & Le (2017); Elsken et al. (2019). Moreover, the neural architectures found via NAS show better performance than the manually-designed networks in many mainstream applications Real et al. (2017); Gong et al. (2019); Xie et al. (2019); Wu et al. (2019); Wan et al. (2020); Li & Talwalkar (2020); Kandasamy et al. (2018); Yu et al. (2020b); Liu et al. (2018b); Cai et al. (2018); Zhang et al. (2019a); Zhou et al. (2019); Howard et al. (2019); Li et al. (2021b).

Despite these advantages, many existing NAS approaches involve a time-consuming and resource-intensive search process. For example, multi-shot NAS uses a controller or an accuracy predictor to conduct the search process and it requires training of multiple networks; thus, multi-shot NAS is extremely time-consuming Real et al. (2019); Chiang et al. (2019). Alternatively, one-shot NAS merges all possible networks from the search space into a supernet and thus only needs to train the supernet once Dong & Yang (2019); Zela et al. (2020); Chen et al. (2019); Cai et al. (2019); Stamoulis et al. (2019); Chu et al. (2021); Guo et al. (2020); Li et al. (2020); this enables one-shot NAS to find a good architecture with much less search time. Though the one-shot NAS has significantly improved the time efficiency of NAS, training is still required during the search process.

---

*Work done while Kartikeya Bhardwaj was at Arm, Inc.

Recently, the *zero-shot* approaches have been proposed to liberate NAS from training entirely Wu et al. (2021); Zhou et al. (2022; 2020); Ingolfsson et al. (2022); Tran & Bae (2021); Do & Luong (2021); Tran et al. (2021); Shu et al. (2022b); Li et al. (2022). Essentially, the zero-shot NAS utilizes some proxy that can predict the test performance of a given network *without training*. The design of such proxies is usually based on some theoretical analysis of deep networks. For instance, the first zero-shot proxy called NN-Mass was proposed by Bhardwaj et al. (2019); NN-Mass theoretically links how the network topology influences gradient propagation and model performance. Hence, zero-shot approaches can not only significantly improve the time efficiency of NAS, but also deepen the theoretical understanding on why certain networks work well. While NN-Mass consistently outperforms #Params, it is not defined for generic NAS topologies and works mostly for simple repeating blocks like ResNets/MobileNets/DenseNets Bhardwaj et al. (2019). Later several zero-shot proxies are proposed for general neural architectures. Nonetheless, as revealed in Ning et al. (2021); White et al. (2022), these general zero-shot proxies proposed to date cannot consistently work better than a naive proxy, namely, the number of parameters (#Params). These results may undermine the effectiveness of zero-shot NAS approaches.

To address the limitations of existing zero-shot proxies, we target the following **key questions**:

1. How do some specific gradient properties, i.e., mean value and standard deviation across different samples, impact the training convergence of neural networks?
2. Can we use these two gradient properties to design a new theoretically-grounded proxy that works better than #Params consistently across many different NAS topologies/tasks?

To this end, we first analyze how the mean value and standard deviation of gradients across different training batches impact the training convergence of neural networks. Based on our analysis, we propose *ZiCo*, a new proxy for zero-shot NAS. We demonstrate that, compared to all existing proxies (including #Params), ZiCo has either a higher or at least on-par correlation with the test accuracy on popular NAS-Benchmarks (NASBench101, NATS-Bench-SSS/TSS) for multiple datasets (CIFAR10/100, ImageNet16-120). Finally, we demonstrate that ZiCo enables a zero-shot NAS framework that can efficiently find the network architectures with the highest test accuracy compared to other zero-shot baselines. In fact, our zero-shot NAS framework achieves competitive FLOPs-accuracy tradeoffs compared to multiple one-shot and multi-shot NAS, but with much lower time costs. To summarize, we make the following **major contributions**:

- We theoretically reveal how the mean value and variance of gradients across multiple samples impact the training convergence and generalization capacity of neural networks.
- We propose a new zero-shot proxy, ZiCo, that works better than existing proxies on popular NAS-Benchmarks (NASBench101, NATS-Bench-SSS/TSS, TransNASBench-101) for multiple applications (image classification/reconstruction and pixel-level prediction).
- We demonstrate that our proposed zero-shot NAS achieves competitive test accuracy with representative one-shot and multi-shot NAS with much less search time.

The rest of the paper is organized as follows. We discuss related work in Section 2. In Section 3, we introduce our theoretical analysis. We introduce our proposed zero-shot proxy (ZiCo) and the NAS framework in Section 3.4. Section 4 validates our analysis and presents our results with the proposed zero-shot NAS. We conclude the paper in Section 5 with remarks on our main contribution.

## 2 RELATED WORK

### 2.1 ZERO-SHOT NAS

The goal of zero-shot NAS is to rank the accuracy of various candidate network architectures *without training*, such that we can replace the expensive training process in NAS with some computation-efficient proxies Xiang et al. (2021a); Javaheripi et al. (2022); Li et al. (2021a). Hence, the quality of the proxy determines the effectiveness of zero-shot NAS. Several works use the number of linear regions to approximately measure the expressivity of a deep neural network Mellor et al. (2021); Chen et al. (2021b); Bhardwaj et al. (2022). Alternatively, most of the existing proxies are derived from the gradient of deep networks. For example, Synflow, SNIP, and GraSP rely on the gradient w.r.t the parameters of neural networks; they are proved to be the different approximations of Taylor expansion of deep neural networks Abdelfattah et al. (2021); Lee et al. (2019b); Tanaka et al. (2020); Wang et al. (2020). Moreover, the Zen-score approximates the gradient w.r.t featuremaps and measures the complexity of neural networks Lin et al. (2021); Sun et al. (2021). Furthermore, Jacob_cov leverages the Jacobian matrix between the loss and multiple input samples to quantify the capacity

of modeling the complex functions Lopes et al. (2021). Though zero-shot NAS can significantly accelerate the NAS process, it has been revealed that the naive proxy #Params generally works better than all the proxies proposed to date Ning et al. (2021); White et al. (2022). These limitations of existing proxies motivate us to look for a new proxy that can consistently work better than #Params and address the limitations of existing zero-shot NAS approaches.

## 2.2 KERNEL METHODS IN NEURAL NETWORKS

Kernel methods are widely explored to analyze the convergence property and generalization capacity of networks trained with gradient descent Neal (1996); Williams (1996); Du et al. (2019a); Lu et al. (2020); Allen-Zhu et al. (2019); Hanin & Nica (2020); Golikov et al. (2022). For example, the training of wide neural networks is proved to be equivalent to the optimization of a specific kernel function Arora et al. (2019a); Lee et al. (2019a); Chizat et al. (2019); Arora et al. (2019b); Cho & Saul (2009). Moreover, given the networks with specific width constraints, researchers proved that the training convergence and generalization capacity of networks can be described by some corresponding kernels Mei et al. (2019); Zhang et al. (2019b); Garriga-Alonso et al. (2019); Du et al. (2019b). In our work, we extend such kernel-based analysis to reveal the relationships between the gradient properties and the training convergence and generalization capacity of neural networks.

## 3 CONVERGENCE AND GENERALIZATION VIA GRADIENT ANALYSIS

We consider the mean value and standard deviation of gradients across different samples and first explore how these two metrics impact the training convergence of linear regression tasks.

### 3.1 LINEAR REGRESSION

Inspired by Du et al. (2019b), we use the training set $\mathbb{S}$ with $M$ samples as follows:

$$\mathbb{S} = \{(\boldsymbol{x_i}, y_i), i = 1, ..., M, \ \boldsymbol{x_i} \in \mathbb{R}^d, \ y_i \in \mathbb{R}, \ ||\boldsymbol{x_i}|| = 1, \ |y_i| \leq R, \ M > 1\} \quad (1)$$

where $R$ is a positive constant and $||\cdot||$ denotes the $L2$-norm of a given vector; $\boldsymbol{x_i} \in \mathbb{R}^d$ is the $i^{th}$ input sample and normalized by its $L2$-norm (i.e., $||\boldsymbol{x_i}|| = 1$), and $y_i$ is the corresponding label. We define the following linear model $f = \boldsymbol{a}^T \boldsymbol{x}$ optimized with an MSE-based loss function $L$:

$$min_{\boldsymbol{a}} \sum_i L(y_i, f(\boldsymbol{x_i}; \boldsymbol{a})) = min_{\boldsymbol{a}} \sum_i \frac{1}{2}(\boldsymbol{a}^T \boldsymbol{x_i} - y_i)^2 \quad (2)$$

where $\boldsymbol{a} \in \mathbb{R}^d$ is the initial weight vector of $f$. We denote the gradient of $L$ w.r.t to $\boldsymbol{a}$ as $g(\boldsymbol{x_i})$ when taking $(\boldsymbol{x_i}, y_i)$ as the training sample:

$$g(\boldsymbol{x_i}) = \frac{\partial L(y_i, f(\boldsymbol{x_i}; \boldsymbol{a}))}{\partial \boldsymbol{a}} \quad (3)$$

We denote the $j^{th}$ element of $g(\boldsymbol{x_i})$ as $g_j(\boldsymbol{x_i})$. We compute the mean value ($\mu_j$) and standard deviation ($\sigma_j$) of $g_j(\boldsymbol{x_i})$ across all training samples as follows:

$$\mu_j = \frac{1}{M} \sum_i^M g_j(\boldsymbol{x_i}) \qquad \sigma_j = \sqrt{\frac{1}{M} \sum_i^M (g_j(\boldsymbol{x_i}) - \mu_j)^2} \quad (4)$$

**Theorem 3.1.** *We denote the updated weight vector as $\hat{\boldsymbol{a}}$ and denote $\sum_{ij}[g_j(\boldsymbol{x_i})]^2 = G$. Assume we use the accumulated gradient of all training samples and learning rate $\eta$ to update the initial weight vector $\boldsymbol{a}$, i.e., $\hat{\boldsymbol{a}} = \boldsymbol{a} - \eta \sum_i g(\boldsymbol{x_i})$. If the learning rate $0 < \eta < 2$, then the total training loss is bounded as follows:*

$$\sum_i L(y_i, f(\boldsymbol{x_i}; \hat{\boldsymbol{a}})) \leq \frac{G}{2} - \frac{\eta}{2}M^2(2 - \eta) \sum_j \mu_j^2 \quad (5)$$

*In particular, if the learning rate $\eta = \frac{1}{M}$, then $L(\hat{\boldsymbol{a}})$ is bounded by:*

$$\sum_i L(y_i, f(\boldsymbol{x_i}; \hat{\boldsymbol{a}})) \leq \frac{M}{2} \sum_j \sigma_j^2 \quad (6)$$

We provide the proof in Appendix A and the experimental results to validate this theorem in Sec 4.2.

***Remark 3.1*** Intuitively, Theorem. 3.1 tells us that the higher the gradient absolute mean across different training samples, the lower the training loss the model converges to; i.e., the network converges at a faster rate. Similarly, if $\eta M < 1$, the smaller the gradient standard deviation across different training samples/batches, the lower the training loss the model can achieve.

## 3.2 MLPs with ReLU

In this section, we generalize the linear model to a network with ReLU activation functions. We primarily consider the standard deviation of gradients in the Gaussian kernel space. We still focus on the regression task on the training set $\mathbb{S}$ defined in Eq. 1. We consider a neural network in the same form as Du et al. (2019b):

$$h(\boldsymbol{x}; \boldsymbol{s}, \boldsymbol{W}) = \frac{1}{\sqrt{m}} \sum_{i}^{m} s_r \text{ReLU}(\boldsymbol{w_r}^T \boldsymbol{x}) \tag{7}$$

where $m$ is the number of output neurons of the first layer; $s_r$ is the $r^{th}$ element in the output weight vector $\boldsymbol{s}$; $\boldsymbol{W} \in \mathbb{R}^{m \times d}$ is the input weight matrix, and $\boldsymbol{w_r} \in \mathbb{R}^d$ is the $r^{th}$ row weight vector in $\boldsymbol{W}$.

For training on the dataset $\mathbb{S}$ with $M$ samples defined in Eq. 1, we minimize the following loss function:

$$L(\boldsymbol{s}, \boldsymbol{W}) = \sum_{i=1}^{M} \frac{1}{2}(h(\boldsymbol{x_i}; \boldsymbol{s}, \boldsymbol{W}) - y_i)^2 \tag{8}$$

Following the common practice Du et al. (2019b), we fix the second layer ($\boldsymbol{s}$) and use gradient descent to optimize the first layer ($\boldsymbol{W}$) with a learning rate $\eta$:

$$\boldsymbol{w_r}(t) = \boldsymbol{w_r}(t-1) - \eta \sum_{i=0}^{t} \frac{\partial L(\boldsymbol{s}, \boldsymbol{W}(t-1))}{\partial \boldsymbol{w_r}(t-1)} \tag{9}$$

where $\boldsymbol{W}(t-1)$ denote the input weight matrix after $t-1$ training steps; $\boldsymbol{w_r}(t)$ denote the $r^{th}$ row weight vector after $t$ training steps.

**Definition 1.** *(Gram Matrix) A Gram Matrix $H(t) \in \mathbb{R}^{M \times M}$ on the training set $\{(\boldsymbol{x_i}, y_i), i = 1, ..., M\}$ after $t$ training steps is defined as follows:*

$$H_{ij}(t) = \frac{1}{m} \boldsymbol{x_i}^T x_j \sum_{r=1}^{m} \mathbb{I}\{\boldsymbol{x_i}^T \boldsymbol{w_r}(t) \geq 0, x_j^T \boldsymbol{w_r}(t) \geq 0\} \tag{10}$$

where $\mathbb{I}$ is the indicator function and $\mathbb{I}\{\mathcal{A}\} = 1$ if and only if event $\mathcal{A}$ happens. We denote the $\lambda_{min}(H)$ as the minimal eigenvalue of a given matrix $H$. We denote the $\lambda_0 = \lambda_{min}(H(\infty))$.

### 3.2.1 Convergence Rate

**Theorem 3.2.** *Given a neural network with ReLU activation function optimized by minimizing Eq. 8, we assume that each initial weight vector $\{\boldsymbol{w_r}(0), r = 1, ..., n\}$ is i.i.d. generated from $\mathcal{N}(0, I)$ and the gradient for each weight follows i.i.d. $\mathcal{N}(0, \sigma)$, where the $\sigma$ is measured across different training steps. For some positive constants $\delta$ and $\epsilon$, if the learning rate $\eta$ satisfies $\eta < \frac{\lambda_0 \sqrt{\pi}\delta}{2M^2\sqrt{2}\Phi(1-\epsilon)t\sigma}$, then with with probability at least $(1-\delta)(1-\epsilon)$, the following holds true: for any $r \in [m]$, $||\boldsymbol{w_r}(0) - \boldsymbol{w_r}(t)|| \leq C = \eta t \sigma \sqrt{\Phi(1-\epsilon)}$, and at training step $t$ the Gram matrix $H(t)$ satisfies:*

$$\lambda_{min}(H(t)) \geq \lambda_{min}(H(0)) - \frac{2\sqrt{2}M^2\eta t\sigma\sqrt{\Phi(1-\epsilon)}}{\sqrt{\pi}\delta} > 0 \tag{11}$$

$\Phi(\cdot)$ *is the inverse cumulative distribution function for a $d$-degree chi-squared distribution $\chi^2(d)$.*

We provide the proof in Appendix B. We now introduce the following result from Du et al. (2019b) to further help our analysis.

**Lemma 1.** *Du et al. (2019b) Assume we set the number of output neurons of the first layer $m = \Omega(\frac{M^6}{\lambda_0^4 \delta^3})$ and we i.i.d. initialize $\boldsymbol{w_r} \sim \mathcal{N}(0, I)$ and $s_r \sim uniform[\{-1, 1\}]$, for $r \in [m]$. When minimizing the loss function in Eq. 8 on the training set $\mathbb{S}$ in Eq. 1, with probability at least $1 - \delta$ over the initialization, the training loss after $t$ training steps is bounded by:*

$$L(\boldsymbol{s}, (\boldsymbol{W}(t)) \leq e^{-\lambda_{min}(H(t))} L(\boldsymbol{s}, (\boldsymbol{W}(t-1)) \tag{12}$$

**Theorem 3.3.** *Under the assumptions of Theorem 3.2 and Lemma 1, with probability at least $(1 - \delta)(1 - \epsilon)$, the following inequality holds true:*

$$L(\boldsymbol{s}, (\boldsymbol{W}(t)) \leq e^{-\lambda_{min}(H(0))} e^{\frac{2\sqrt{2}M^2\eta t\sigma\sqrt{\Phi(1-\epsilon)}}{\sqrt{\pi}\delta}} L(\boldsymbol{s}, (\boldsymbol{W}(t-1)) \tag{13}$$

The proof consists of replacing $\lambda_{min}(H(t))$ in Eq. 12 with its lower bound given by Theorem 3.2.

**Remark 3.4** Theorem. 3.3 shows that after some training steps $t$, the network with a smaller standard deviation ($\sigma$) of gradients will have a smaller training loss; i.e., the network has a faster convergence rate at each training step. We further validate this theorem in Sec. 4.2.

### 3.2.2  GENERALIZATION CAPACITY

Several prior works show that the generalization capacity of a neural network is highly correlated with its sharpness of the loss function Keskar et al. (2017b;a); Li et al. (2018); Liang et al. (2019). Usually, a flatter loss landscape leads to a better generalization capacity. Moreover, it has also been shown that the largest eigenvalue of the Gram matrix of loss can be used to describe the sharpness of the loss landscape Sagun et al. (2018); more precisely:

**Proposition 3.4.** *The lower the largest eigenvalue of the Gram matrix, the higher the generalization capacity of the network. [Lewkowycz et al. (2020); Sagun et al. (2016)]*

Next, we analyze how the gradient of a neural network impacts the largest eigenvalues of the Gram matrix and its generalization capacity.

**Theorem 3.5.** *Under the assumptions of Theorem 3.2, for some positive constants $\delta$ and $\epsilon$, if the learning rate $\eta$ satisfies $\eta < \frac{\lambda_0\sqrt{\pi}\delta}{2M^2\sqrt{2}\Phi(1-\epsilon)t\sigma}$, then with with probability at least $(1-\delta)(1-\epsilon)$, for any $r \in [m]$, $\|\boldsymbol{w_r}(0) - \boldsymbol{w_r}(t)\| \leq C = \eta t\sigma\sqrt{\Phi(1-\epsilon)}$, and at training step $t$, the Gram matrix $H(t)$ satisfies:*

$$\lambda_{max}(H(t)) \leq \lambda_{max}(H(0)) + \frac{2\sqrt{2}M^2\eta t\sigma\sqrt{\Phi(1-\epsilon)}}{\sqrt{\pi}\delta} \tag{14}$$

$\Phi(\cdot)$ *is the inverse cumulative distribution function for a $d$-degree chi-squared distribution $\chi^2(d)$.*

We provide the proof in Appendix C.

**Remark 3.5** Theorem. 3.5 shows that after some training steps $t$, the network with a smaller standard deviation ($\sigma$) of gradients will have a lower largest eigenvalues of the Gram matrix; i.e., the network has a flatter loss landscape rate at each training step. Therefore, based on Proposition 3.4, the model will generalize better. We further validate this theorem in the following section.

### 3.3  SUMMARY OF OUR THEORETICAL ANALYSIS

Theorem 3.1, Theorem 3.3 and Theorem 3.5 tell us that the network with a high training convergence speed and generalization capacity should have *high absolute mean values* and *low standard deviation values* for the gradient, w.r.t the parameters across different training samples/batches.

### 3.4  NEW ZERO-SHOT PROXY

Inspired by the above theoretical insights, we next propose a proxy (ZiCo) that jointly considers both absolute mean and standard deviation values. Following the standard practice, we consider convolutional neural networks (CNNs) as candidate networks.

**Definition 2.** *Given a neural network with $D$ layers and loss function $L$, the Zero-shot inverse Coefficient of Variation (ZiCo) is defined as follows:*

$$ZiCo = \sum_{l=1}^{D} log\left(\sum_{\omega \in \boldsymbol{\theta}_l} \frac{\mathbb{E}[|\nabla_\omega L(\boldsymbol{X_i}, \boldsymbol{y_i}; \Theta)|]}{\sqrt{Var(|\nabla_\omega L(\boldsymbol{X_i}, \boldsymbol{y_i}; \Theta)|)}}\right), \quad i \in \{1, ..., N\} \tag{15}$$

where $\Theta$ denote the initial parameters of a given network; $\boldsymbol{\theta}_l$ denote the parameters of the $l^{th}$ layer of the network, and $\omega$ represents each element in $\boldsymbol{\theta}_l$; $\boldsymbol{X_i}$ and $\boldsymbol{y_i}$ are the $i^{th}$ input batch and corresponding labels from the training set; $N$ is number of training batches used to compute ZiCo. We incorporate $log$ to stabilize the computation by regularizing the extremely large or small values.

Of note, our metric is applicable to general CNNs; i.e., there's no restriction w.r.t. the neural architecture when calculating ZiCo. As discussed in Section 3.3, the networks with higher ZiCo tend

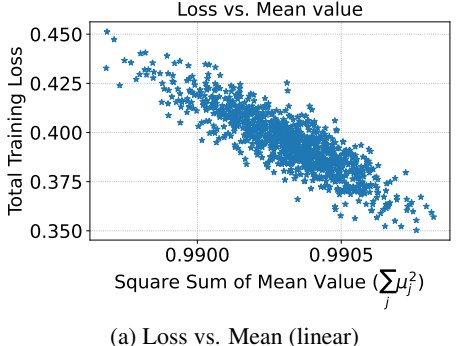
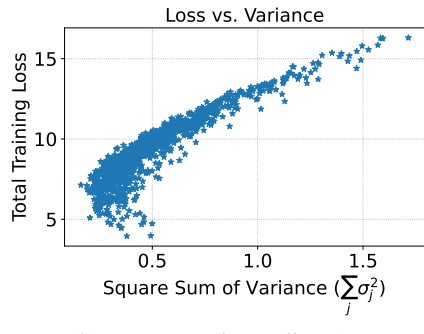

(a) Loss vs. Mean (linear)  (b) Loss vs. variance (linear)

Figure 1: Training loss vs. square sum of mean gradients and the sum of gradients variances for linear networks on MNIST after one epoch. Clearly, larger mean gradient values lead to lower loss values; also, networks with smaller $\sum_j \sigma_j^2$ have lower loss values.

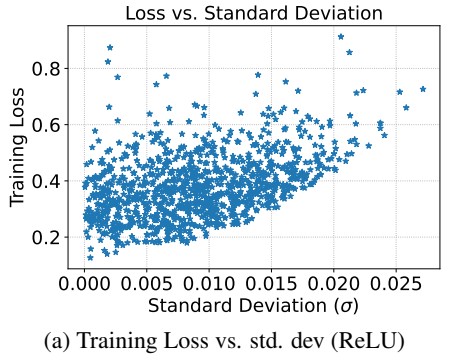
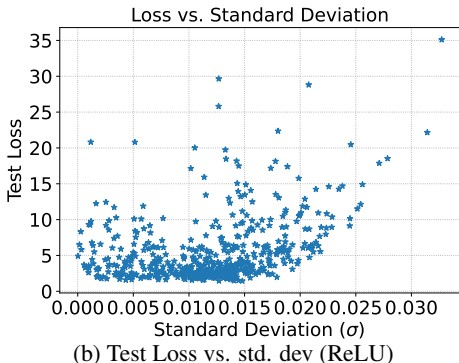

(a) Training Loss vs. std. dev (ReLU)  (b) Test Loss vs. std. dev (ReLU)

Figure 2: Training loss and Test loss vs. standard deviation of gradients for two-layer MLPs with ReLU on MNIST after one training epoch. Networks with smaller $\sigma$ tend to have lower training loss and test loss values. We provide more results in Appendix C.1.

to have better convergence rates and higher generalization capacity. Hence, the architectures with higher ZiCo are better architectures.

We remark that the loss values in Eq. 15 are all computed with the **initial** parameters $\Theta$; that is, we **never** update the value of the parameters when computing ZiCo for a given network (hence it follows the basic principle of zero-shot NAS, i.e., never train, only use the initial parameters). In practice, two batches are enough to make ZiCo achieve the SOTA performance among all previously proposed accuracy proxies (see Sec. 4.5). Hence, we use only two input batches ($N = 2$) to compute ZiCo; this makes ZiCo very time efficient for a given network.

## 4 EXPERIMENTAL RESULTS

### 4.1 EXPERIMENTAL SETUP

We conduct the following types of experiments: (*i*) Empirical validation of Theorem 3.1, Theorem 3.3 and Theorem 3.5; (*ii*) Evaluation of the proposed ZiCo on multiple NAS benchmarks; (*iii*) Illustration of ZiCo-based zero-shot NAS on ImageNet.

For the experiments (*i*), to validate Theorem 3.1, we optimize a linear model as in Eq. 2 on the MNIST dataset, the mean gradient values and the standard deviation vs. the total training loss. Moreover, we also optimize the model defined by Eq. 7 on MNIST and report the training loss vs. the standard deviation in order to validate Theorem 3.2 and Theorem 3.5.

For experiments (*ii*), we compare our proposed ZiCo against existing proxies on three mainstream NAS benchmarks: ***NATSBench*** is a popular cell-based search space with two different search spaces: (1) NATSBench-TSS consisting of 15625 total architectures with different cell structures trained on CIFAR10, CIFAR100, and ImageNet16-120 (Img16-120) datasets, which is just renamed

Table 1: The correlation coefficients between various zero-cost proxies and two naive proxies (#Params and FLOPs) vs. test accuracy on NATSBench-TSS (KT and SPR represent Kendall's $\tau$ and Spearman's $\rho$, respectively). The best results are shown with bold fonts. Clearly, ZiCo is the only proxy that works consistently better than #Params and is generally the best proxy. We provide more results in Table 3 and Table 4 in Appendix E.1.

| NATSBench-TSS (NASBench201) | | | | | | |
|---|---|---|---|---|---|---|
| Dataset | CIFAR10 | | CIFAR100 | | Img16-120 | |
| Proxy      Correlation | KT | SPR | KT | SPR | KT | SPR |
| Grad_norm Abdelfattah et al. (2021) | 0.46 | 0.63 | 0.47 | 0.63 | 0.43 | 0.58 |
| SNIP Lee et al. (2019b) | 0.46 | 0.63 | 0.46 | 0.63 | 0.43 | 0.58 |
| GraSP Wang et al. (2020) | 0.37 | 0.54 | 0.36 | 0.51 | 0.40 | 0.56 |
| Fisher Liu et al. (2021) | 0.40 | 0.55 | 0.41 | 0.55 | 0.37 | 0.50 |
| Synflow Tanaka et al. (2020) | 0.54 | 0.73 | 0.57 | 0.76 | 0.56 | 0.75 |
| Zen-score Lin et al. (2021) | 0.29 | 0.38 | 0.28 | 0.36 | 0.29 | 0.40 |
| FLOPs | 0.54 | 0.73 | 0.51 | 0.71 | 0.49 | 0.67 |
| *#Params* | *0.57* | *0.75* | *0.55* | *0.73* | *0.52* | *0.69* |
| **ZiCo (Ours)** | **0.61** | **0.80** | **0.61** | **0.81** | **0.60** | **0.79** |

from NASBench-201 Dong & Yang (2020); (2) NATSBench-SSS contains includes 32768 architectures (which differ only in the width values of each layer) and is also trained on the same three above datasets Dong et al. (2021). ***NASBench101*** provides users with 423k neural architectures with their test accuracy on CIFAR10 dataset; the architectures are built by stacking the same cell multiple times Ying et al. (2019). ***TransNASBench- 101-Mirco*** contains 4096 networks with different cell structures on various downstream applications (see Appendix E.2) Duan et al. (2021).

For experiments (*iii*), we use ZiCo to conduct the zero-shot NAS (see Algorithm 1) on ImageNet. We first use Algorithm 1 to find the networks with the highest ZiCo under various FLOPs budgets. We conduct the search for 100k steps; this takes 10 hours on a single NVIDIA 3090 GPU (i.e., 0.4 GPU days). Then, we train the obtained network with the exact same training setup as Lin et al. (2021). Specifically, we train the neural network for 480 epochs with the batch size 512 and input resolution 224. We also use the distillation-based training loss functions by taking Efficient-B3 as the teacher. Finally, we set the initial learning rate as 0.1 with a cosine annealing scheduling scheme.

## 4.2 VALIDATION OF THEOREMS 3.1&3.3&3.5

To empirically validate Theorem 3.1, we first randomly sample 1000 training images in MNIST; we then normalize these images with their $L2$-norm to create the training set $\mathbb{S}$. We compute the gradient w.r.t. the network parameters for each individual training sample. Next, as discussed in Theorem 3.1, we use the accumulated gradient over these samples to update the network parameters with learning rate $\eta = 1$. Then, we calculate the square sum of mean gradients and the total training loss. We repeat the above process 1000 times on the same $\mathbb{S}$. As shown in Fig. 1(a), we plot the total training loss vs. square sum of mean gradients as defined in Eq. 5. Clearly, the networks with the higher square sum of mean gradients values tend to have lower training loss. In comparison, Fig. 1(b) shows that networks with a lower square sum of variance value tend to have lower training loss values, which coincides with the conclusion drawn from Eq. 6. These results empirically validate our Theorem 3.1.

Moreover, to optimize a two-layer MLP with ReLU activation functions as defined in Eq. 7, we use the entire training set of MNIST and apply the gradient descent (Eq. 9) to update the weights. We set the batch size as 256 and measure the standard deviation of gradients ($\sigma$) w.r.t. parameters across different training batches. We set a very small learning rate $\eta = 10^{-8}$ to satisfy the assumption in Theorem 3.2 and Theorem 3.5. We plot the training loss and test loss after one training epoch vs. standard deviation of gradients ($\sigma$) in Fig. 2(a). Clearly, the results show that if a network has a lower gradient standard deviation, then it tends to have lower training loss values, and thus, a faster convergence rate. These results empirically prove our claims in Theorem 3.3. Similarly, Fig. 2(a) shows that if a network has a lower gradient standard deviation, then it tends to have lower test loss values, which empirically validates Theorem 3.5.

## 4.3 ZICO VS. OTHER PROXIES ON NAS BENCHMARKS

We first calculate the correlation coefficients between various proxies and the test accuracy on CIFAR10, CIFAR100, and ImageNet16-120 datasets for NATSBench-TSS. As shown in Table. 1, ZiCo achieves the highest correlation with the real rest accuracy. We provide more results in Appendix E.

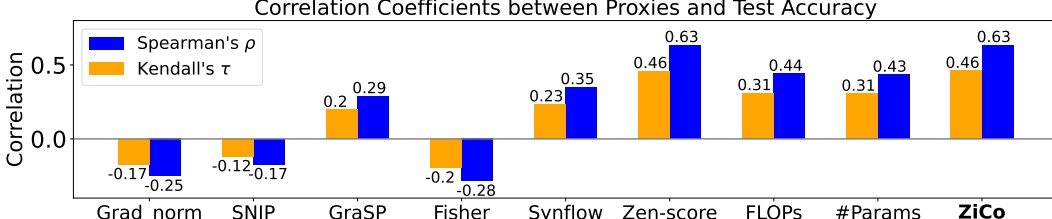

Figure 3: Correlation coefficients of various proxies vs. test accuracy on NASBench101 search space. ZiCo has significantly higher correlation scores than other proxies, except for Zen-score.

Table 2: Comparison of Top-1 accuracy of our ZiCo-based NAS against SOTA NAS methods on ImageNet under various FLOP budgets (averages over three runs). For the 'Method' column, 'MS' means multi-shot NAS; 'OS' is short for one-shot NAS; Scaling represents network scaling methods; 'ZS' is short for zero-shot NAS. OFA[‡] is trained from scratch and reported in Moons et al. (2021).

| Budget (maximal #FLOPs) | Approach | FLOPs | Top-1 [%] | Method | Costs [GPU Days] |
|---|---|---|---|---|---|
| 450M | EfficientNet-B0 Tan & Le (2019) | 390M | 77.1 | Scaling | 3800 |
| | MnasNet-A3 Tan et al. (2019) | 403M | 76.7 | MS | - |
| | OFA[‡] Cai et al. (2020) | 406M | 77.7 | OS | 50 |
| | BN-NAS Chen et al. (2021a) | 470M | 75.7 | MS | 0.8 |
| | NASNet-B Zoph et al. (2018) | 488M | 72.8 | MS | 1800 |
| | CARS-D Yang et al. (2020) | 496M | 73.3 | MS | 0.4 |
| | DONNA Moons et al. (2021) | 501M | 78.0 | OS | 25 |
| | #Params | 451M | 63.5 | ZS | 0.02 |
| | **ZiCo (Ours)** | **448M** | **78.1±0.3** | **ZS** | **0.4** |
| 600M | DARTS Liu et al. (2019) | 574M | 73.3 | OS | 4 |
| | PC-DARTS Xu et al. (2019) | 586M | 75.8 | OS | 3.8 |
| | BigNAS-L Yu et al. (2020a) | 586M | 79.5 | OS | 2304 (TPU days) |
| | CARS-I Yang et al. (2020) | 591M | 75.2 | MS | 0.4 |
| | EnTranNAS Yang et al. (2021) | 594M | 76.2 | OS | 2.1 |
| | MAGIC-AT Xu et al. (2022) | 598M | 76.8 | OS | 2 |
| | SemiNAS Luo et al. (2020) | 599M | 76.5 | MS | 4 |
| | DONNA Moons et al. (2021) | 599M | 78.4 | OS | 25 |
| | Zen-score Lin et al. (2021) | 611M | 79.1 | ZS | 0.5 |
| | OFA[‡] Cai et al. (2020) | 662M | 78.7 | OS | 50 |
| | EfficientNet-B1 Tan & Le (2019) | 700M | 79.1 | Scaling | 3800 |
| | **ZiCo (Ours)** | **603M** | **79.4±0.3** | **ZS** | **0.4** |
| 1000M | sharpDARTS Hundt et al. (2019) | 950M | 76.0 | OS | - |
| | Zen-score Lin et al. (2021) | 934M | 80.8 | ZS | 0.5 |
| | EfficientNet-B2 Tan & Le (2019) | 1000M | 80.1 | Scaling | 3800 |
| | **ZiCo (Ours)** | **1005M** | **80.5±0.2** | **ZS** | **0.4** |

For NASBench101, as shown in Fig. 3, ZiCo has a significantly higher correlation score with the real test accuracy than all the other proxies, except Zen-score. For example, ZiCo has a 0.46 Kendall's $\tau$ score, while #Params is only 0.31. In general, ZiCo has the highest correlation coefficients among all existing proxies for various search spaces and datasets of NATSBench and NASBench101. To our best knowledge, ZiCo is the first proxy that shows a consistently higher correlation coefficient compared to #Params.

The above results validate the effectiveness of our proposed ZiCo; thus, ZiCo can be directly used to search for optimal networks for various budgets. Next, we describe the search results in detail.

## 4.4 ZiCo on ImageNet

**Search Space** We use the commonly used MobileNetv2-based search space where the candidate networks are built by stacking multiple Inverted Bottleneck Blocks (IBNs) with SE modules Sandler et al. (2018); Pham et al. (2018); Lin et al. (2021). As for each IBN, the kernel size of the depthwise convolutional layer is sampled from $\{3, 5, 7\}$ and the expansion ratio is randomly selected from $\{1, 2, 4, 6\}$. We consider ReLU as the activation function. We use standard Kaiming_Init to initialize all linear and convolution layers for every candidate networks He et al. (2015). More details of the search space are given in Appendix D.

We use Algorithm 1 (see Appendix D.2) to search networks under various FLOPs budgets (450M, 600M, and 1000M) within the above search space. As shown in Table 2, ZiCo outperforms most previous NAS approaches by a large margin. For example, when the FLOPs budget is around 450M, ZiCo achieves 78.1% Top-1 accuracy, which is competitive with one of the SOTA NAS methods

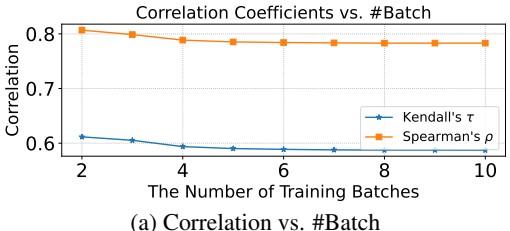 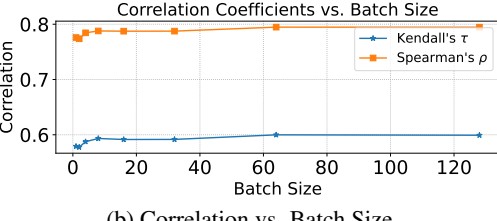

(a) Correlation vs. #Batch          (b) Correlation vs. Batch Size

Figure 4: Ablation study. The correlation coefficients between: (a) ZiCo under varying number of batches and real test accuracy; (b) ZiCo under varying batch size and real test accuracy.

(DONNA), but with fewer FLOPs and 648× faster search speed Moons et al. (2021). Moreover, if the FLOPs is 600M, ZiCo achieves 2.6% higher Top-1 Accuracy than the latest one-shot NAS method (MAGIC-AT) with a 3× reduction in terms of search time Xu et al. (2022).

To make further comparison with #Params, we also use #Params as the proxy and Algorithm 1 to conduct the search under a 450M FLOPs budget. As shown in Table 2, the obtained network by #Params has a 14.6% lower accuracy than ours (63.5% vs. 78.1%). Hence, even though the correlations for ZiCo and #Params in Table 1 and the optimal networks in Table 4 are similar for small-scale datasets, ZiCo significantly outperforms naive baselines like #Params for large datasets like ImageMet. To conclude, ZiCo achieves SOTA results for Zero-Shot NAS and outperforms naive methods, existing zero-shot proxies, as well as several one-shot and multi-shot methods.

We remark that these results demonstrate two benefits of our proposed ZiCo: (*i*) **Lightweight computation costs.** As discussed in Sec 3, during the search process, to evaluate a given architecture, we only need to conduct the backward propagation twice (only takes 0.3s on an NVIDIA 3090 GPU). The computation efficiency and exemption of training enable ZiCo to significantly reduce the search time of NAS. (*ii*) **High correlation with the real test accuracy.** As demonstrated in Sec 4.3, ZiCo has a very high correlation score with real accuracy for architectures from various search spaces and datasets. Hence, ZiCo can accurately predict the test accuracy of diverse neural architectures, thus helping find the optimal architectures with the best test performance.

### 4.5 ABLATION STUDY

**Number of batches** We randomly select 2000 networks from NATSBench-TSS on CIFAR100 dataset and compute ZiCo under varying number of training batches ($N$ in Eq. 15) from $\{2,...,10\}$. We then calculate the correlation between ZiCo with the real test accuracy. Fig. 4(a) shows that using two batches to compute ZiCo generates the highest score. Hence, in our work, we always use two batches ($N = 2$) to compute ZiCo since it is both accurate and time-efficient.

**Batch size** We compute ZiCo with two batches under varying batch size $\{1, 2, 4, 8, 16, 32, 64, 128\}$ for the same 2000 networks as above; we then calculate the correlation between ZiCo with the test accuracy. Fig. 4(b) shows that batch size 64 is enough to stabilize the coefficient. Hence, we set the batch size as 128 and use two batches to compute ZiCo. We provide more ablation studies in Appendix F.

### 5 CONCLUSION

In this work, we have proposed ZiCo, a new SOTA proxy for zero-shot NAS. As the main theoretical contribution, we first reveal how the mean value and standard deviation of gradients impact the training convergence of a given architecture. Based on this theoretical analysis, we have shown that ZiCo works better than all zero-shot NAS proxies proposed so far on multiple popular NAS-Benchmarks (NASBench101, NATSBench-SSS/TSS) for multiple datasets (CIFAR10/100, ImageNet16-120). In particular, we have demonstrated that ZiCo is consistently better than (#Params) and existing zero-shot proxies. Moreover, ZiCo enables us to find architectures with competitive test performance to representative one-shot and multi-shot NAS methods, but with much lower search costs. For example, ZiCo-based NAS can find the architectures with 78.1%, 79.4%, and 80.4% test accuracies under 450M, 600M, and 1000M FLOPs budgets, respectively, on ImageNet within 0.4 GPU days.

### ACKNOWLEDGMENTS

This work was supported in part by the US National Science Foundation (NSF) grant CNS-2007284.

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

# A   PROOF OF THEOREM 3.1

**Theorem 3.1** *We denote the updated weight vector as $\hat{\boldsymbol{a}}$ and $\sum_{ij}[g_j(\boldsymbol{x_i})]^2 = G$. Assume we use the accumulated gradient of all training samples and learning rate $\eta$ to update the initial weight vector $\boldsymbol{a}$, i.e., $\hat{\boldsymbol{a}} = \boldsymbol{a} - \eta \sum_i g(\boldsymbol{x_i})$. If the learning rate $0 < \eta < 2$, then the total training loss is bounded as follows:*

$$\sum_i L(y_i, f(\boldsymbol{x_i}; \hat{\boldsymbol{a}})) \leq \frac{G}{2} - \frac{\eta}{2} M^2 (2 - \eta) \sum_j \mu_j^2 \tag{16}$$

*In particular, if the learning rate $\eta = \frac{1}{M}$, then $L(\hat{\boldsymbol{a}})$ is bounded by:*

$$\sum_i L(y_i, f(\boldsymbol{x_i}; \hat{\boldsymbol{a}})) \leq \frac{M}{2} \sum_j \sigma_j^2 \tag{17}$$

*Proof.* Given each training sample $(\boldsymbol{x_i}, y_i)$ the gradient of $L$ w.r.t to $\boldsymbol{a}$ when taking $(\boldsymbol{x_i}, y_i)$ as the input is as follows:

$$g(\boldsymbol{x_i}) = \frac{\partial L(y_i, f(\boldsymbol{x_i}; \boldsymbol{a}))}{\partial a} = \boldsymbol{x_i} \boldsymbol{x_i}^T \boldsymbol{a} - y_i \boldsymbol{x_i} \tag{18}$$

We note that:

$$\begin{aligned}
(\boldsymbol{a} - g(\boldsymbol{x_i}))^T \boldsymbol{x_i} - y_i &= \boldsymbol{a}^T \boldsymbol{x_i} - \boldsymbol{a}^T \boldsymbol{x_i} \boldsymbol{x_i}^T \boldsymbol{x_i} + y_i \boldsymbol{x_i}^T \boldsymbol{x_i} - y_i \\
&= \boldsymbol{a}^T \boldsymbol{x_i} - (\boldsymbol{a}^T \boldsymbol{x_i})(\boldsymbol{x_i}^T \boldsymbol{x_i}) \\
&= \boldsymbol{a}^T \boldsymbol{x_i} - \boldsymbol{a}^T \boldsymbol{x_i} \\
&= 0 \implies y_i = (\boldsymbol{a} - g(\boldsymbol{x_i}))^T \boldsymbol{x_i}
\end{aligned} \tag{19}$$

Then the total training loss among all training samples is given by:

$$\sum_{i=1}^M \frac{1}{2} (\hat{\boldsymbol{a}}^T \boldsymbol{x_i} - y_i)^2 \tag{20}$$

By using Eq. 19, we can rewrite Eq. 20 as follows:

$$\begin{aligned}
\sum_{i=1}^M \frac{1}{2} (\hat{\boldsymbol{a}}^T \boldsymbol{x_i} - y_i)^2 &= \sum_{i=1}^M \frac{1}{2} (\hat{\boldsymbol{a}}^T \boldsymbol{x_i} - (\boldsymbol{a} - g(\boldsymbol{x_i}))^T \boldsymbol{x_i})^2 \\
&= \sum_{i=1}^M \frac{1}{2} ((\hat{\boldsymbol{a}} - \boldsymbol{a} + g(\boldsymbol{x_i}))^T \boldsymbol{x_i})^2
\end{aligned} \tag{21}$$

Recall the assumption that $\hat{\boldsymbol{a}} = \boldsymbol{a} - \eta \sum_i g(\boldsymbol{x_i})$; we rewrite Eq. 21 as follows:

$$\sum_{i=1}^M \frac{1}{2} (\hat{\boldsymbol{a}}^T \boldsymbol{x_i} - y_i)^2 = \sum_{i=1}^M \frac{1}{2} (g(\boldsymbol{x_i}) - \eta \sum_i g(\boldsymbol{x_i}))^T \boldsymbol{x_i})^2 \tag{22}$$

According to the Cauchy–Schwarz inequality and $||x_i|| = 1$, the total training loss is bounded by:

$$
\begin{aligned}
\sum_{i=1}^{M} \frac{1}{2}(\hat{a}^T x_i - y_i)^2 &\leq \frac{1}{2} \sum_{i=1}^{M} ||(g(x_i) - \eta \sum_i g(x_i)||^2 * ||x_i||^2 \\
&= \frac{1}{2} \sum_{i=1}^{M} ||(g(x_i) - \eta \sum_i g(x_i)||^2 \\
&= \frac{1}{2} \sum_{i=1}^{M} \sum_{j=1}^{d} ((g_j(x_i) - \eta M \mu_j)^2 \\
&= \frac{1}{2} \sum_{i=1}^{M} \sum_{j=1}^{d} ([g_j(x_i)]^2 - 2\eta M \mu_j g_j(x_i) + \eta^2 M^2 \mu_j^2) \\
&= \frac{1}{2} \sum_{ij} [g_j(x_i)]^2 + \sum_j \eta^2 M^2 \mu_j^2 - 2 \sum_j (\eta M \mu_j \sum_i g_j(x_i)) \\
&= \frac{1}{2} \sum_{ij} [g_j(x_i)]^2 + \sum_j \eta^2 M^2 \mu_j^2 - 2 \sum_j (\eta M \mu_j M \mu_j) \\
&= \frac{1}{2} G + \sum_j (\eta^2 M^2 \mu_j^2 - 2\eta M^2 \mu_j^2) \\
&= \frac{1}{2} G - \eta M^2 (2 - \eta) \sum_j \mu_j^2
\end{aligned}
\tag{23}
$$

Since $\sum_{i=1}^{M} \frac{1}{2}(\hat{a}^T x_i - y_i)^2$ is always non-negative, the above upper bound of training loss satisfies:

$$
\frac{1}{2} G - \eta M^2 (2 - \eta) \sum_j \mu_j^2 \geq \sum_{i=1}^{M} \frac{1}{2}(\hat{a}^T x_i - y_i)^2 \geq 0
\tag{24}
$$

Note that, if $0 < \eta < 2$, then $\eta(2 - \eta) > 0$. Therefore, the larger $\sum_j \mu_j^2$ term would make the upper bound of training loss in Eq. 23 closer to 0. In other words, the higher the gradient absolute mean values across different training samples/batches, the lower the training loss values the model converges to; i.e., the network converges at a faster rate.

In particular, if $\eta = \frac{1}{M}$, the Eq. 23 can be rewritten as:

$$
\begin{aligned}
\sum_{i=1}^{M} \frac{1}{2}(\hat{a}^T x_i - y_i)^2 &\leq \frac{1}{2} \sum_{i=1}^{M} \sum_{j=1}^{d} ((g_j(x_i) - \mu_j)^2 \\
&= \frac{1}{2} \sum_j M \sigma_j^2 \\
&= \frac{M}{2} \sum_j \sigma_j^2
\end{aligned}
\tag{25}
$$

This completes our proof. $\qquad\square$

## B    PROOF OF THEOREM 3.2

**Theorem 3.2** *Given a neural network with ReLU activation function optimized by minimizing Eq. 8, we assume that each initial weight vector $\{w_r(0), r = 1, ..., n\}$ is i.i.d. generated from $\mathcal{N}(0, I)$ and the gradient for each weight follows an i.i.d. $\mathcal{N}(0, \sigma)$. For some positive constants $\delta$ and $\epsilon$, if the learning rate $\eta$ satisfies $\eta < \frac{\lambda_0 \sqrt{\pi} \delta}{2M^2 \sqrt{2} \Phi(1-\epsilon) t \sigma}$, then with with probability at least $(1 - \delta)(1 - \epsilon)$, the following holds true: for any $r \in [m]$, $||w_r(0) - w_r(t)|| \leq C = \eta t \sigma \sqrt{\Phi(1 - \epsilon)}$, and at training step $t$ the Gram matrix $H(t)$ satisfies:*

$$\lambda_{min}(H(t)) \geq \lambda_{min}(H(0)) - \frac{2\sqrt{2}M^2\eta t\sigma\sqrt{\Phi(1-\epsilon)}}{\sqrt{\pi}\delta} > 0 \tag{26}$$

$\Phi(\cdot)$ *is the inverse cumulative distribution function for a* $d$-*degree chi-squared distribution* $\chi^2(d)$.

*Proof.* We first compute the probability of $||\boldsymbol{w_r}(0) - \boldsymbol{w_r}(t)|| \leq C$. Based on the assumption $w_i(0), i = 1, ..., n\}$ follows i.i.d. $\mathcal{N}(0, I)$ and the gradient for each weight follows i.i.d. $\mathcal{N}(0, \sigma)$, considering the weight updating rule defined in Eq. 9, each element in $\boldsymbol{w_r}(0) - \boldsymbol{w_r}(t)$ follows a i.i.d. $\mathcal{N}(0, \eta t\sigma)$. Therefore, $\frac{||\boldsymbol{w_r}(0) - \boldsymbol{w_r}||^2}{\eta^2 t^2 \sigma^2}$ follows the chi-squared distribution with $d$ degrees of freedom $\chi^2(d)$.

$$
\begin{aligned}
P(||\boldsymbol{w_r}(0) - \boldsymbol{w_r}|| \leq C) &= P(||\boldsymbol{w_r}(0) - \boldsymbol{w_r}(t)||^2 \leq C^2) \\
&= P(\frac{||\boldsymbol{w_r}(0) - \boldsymbol{w_r}(t)||^2}{\eta^2 t^2 \sigma^2} \leq \frac{C^2}{\eta^2 t^2 \sigma^2}) \\
&= P(\frac{||\boldsymbol{w_r}(0) - \boldsymbol{w_r}(t)||^2}{\eta^2 t^2 \sigma^2} \leq \Phi(1-\epsilon)) \\
&= 1 - \epsilon
\end{aligned}
\tag{27}
$$

Given an input sample $\boldsymbol{x}_i$ and a weight vector $\boldsymbol{w_r}(t)$ from $\boldsymbol{W}(t)$, we define the following event:

$$\mathcal{A}_{ir} = \{||\boldsymbol{w_r}(t) - \boldsymbol{w_r}(0)|| \leq C\} \cap \{\mathbb{I}\{\boldsymbol{x_i^T}\boldsymbol{w_r}(0) \geq 0\} \neq \mathbb{I}\{\boldsymbol{x_i^T}\boldsymbol{w_r}(t) \geq 0\}\} \tag{28}$$

If $||\boldsymbol{w_r}(t) - \boldsymbol{w_r}(0)|| \leq C$ holds true,

$$
\begin{aligned}
\boldsymbol{x_i^T}\boldsymbol{w_r}(t) &= \boldsymbol{x_i^T}(\boldsymbol{w_r}(t) - \boldsymbol{w_r}(0)) + \boldsymbol{x_i^T}\boldsymbol{w_r}(0) \\
&= \text{sign}(\boldsymbol{x_i^T}(\boldsymbol{w_r}(t) - \boldsymbol{w_r}(0)))||\boldsymbol{w_r}(t) - \boldsymbol{w_r}(0)|| + \text{sign}(\boldsymbol{x_i^T}\boldsymbol{w_r}(0))||\boldsymbol{w_r}(0)||
\end{aligned}
\tag{29}
$$

Eq. 29 tells us that if $||\boldsymbol{w_r}(0)||$ is larger than $||\boldsymbol{w_r}(t) - \boldsymbol{w_r}(0)||$, then $\boldsymbol{x_i^T}\boldsymbol{w_r}(0)$ determines the sign value of $\boldsymbol{x_i^T}\boldsymbol{w_r}(t)$; in other words, $\boldsymbol{x_i^T}\boldsymbol{w_r}(t)$ always has the same sign values with $\boldsymbol{x_i^T}\boldsymbol{w_r}(0)$; i.e., $\mathbb{I}\{\boldsymbol{x_i^T}\boldsymbol{w_r}(0) \geq 0\} = \mathbb{I}\{\boldsymbol{x_i^T}\boldsymbol{w_r}(t) \geq 0\}$. That is, if $||\boldsymbol{w_r}(t) - \boldsymbol{w_r}(0)|| \leq C$ and $\mathbb{I}\{\boldsymbol{x_i^T}\boldsymbol{w_r}(0) \geq 0\} \neq \mathbb{I}\{\boldsymbol{x_i^T}\boldsymbol{w_r}(t) \geq 0\}$ hold true, then $||\boldsymbol{w_r}(0)|| \leq C$. Therefore, the probability of event $\mathcal{A}_{ir}$:

$$P(\mathcal{A}_{ir}) \leq P(\{||\boldsymbol{w_r}(0)|| \leq C\}) \tag{30}$$

By anti-concentration inequality of Gaussian distribution Du et al. (2019b), we have:

$$P(\mathcal{A}_{ir}) \leq P(\{||\boldsymbol{w_r}(0)|| \leq C\}) \leq \frac{\sqrt{2}C}{\sqrt{\pi}} \tag{31}$$

Therefore, if any weight vector $w_1, ..., w_m$ satisfies $||\boldsymbol{w_r}(0) - \boldsymbol{w_r}(t)|| \leq C$, we can bound the entry-wise deviation on the Gram matrix $H(t)$ at training step $t$: for any $(i, j) \in [n] \times [n]$:

$$
\begin{aligned}
&\mathbb{E}[|H_{ij}(0) - H_{ij}(t)|] \\
=&\mathbb{E}[\frac{1}{m}|\boldsymbol{x_i^T}x_j \sum_{r=1}^{m}(\mathbb{I}\{\boldsymbol{x_i^T}\boldsymbol{w_r}(0) \geq 0, x_j^T\boldsymbol{w_r}(0) \geq 0\} - \mathbb{I}\{\boldsymbol{x_i^T}\boldsymbol{w_r}(t) \geq 0, x_j^T\boldsymbol{w_r}(t) \geq 0\})|] \\
=&\mathbb{E}[\frac{1}{m}|\boldsymbol{x_i^T}x_j \sum_{r=1}^{m}(\mathbb{I}\{\boldsymbol{x_i^T}\boldsymbol{w_r}(0) \geq 0\}\mathbb{I}\{x_j^T\boldsymbol{w_r}(0) \geq 0\} - \mathbb{I}\{\boldsymbol{x_i^T}\boldsymbol{w_r}(t) \geq 0\}\mathbb{I}\{x_j^T\boldsymbol{w_r}(t) \geq 0\})|] \\
\leq&\mathbb{E}[\frac{1}{m}\sum_{r=1}^{m}(\mathbb{I}\{\mathcal{A}_{ir} \cup \mathcal{A}_{jr}\}] \leq P(\mathcal{A}_{ir}) + P(\mathcal{A}_{jr}) \\
\leq&\frac{2\sqrt{2}C}{\sqrt{\pi}}
\end{aligned}
\tag{32}
$$

where the expectation is summing over the initial weight $w(0)$. Hence, considering all the elements in $H$, we have:

$$\mathbb{E}[\sum_{i=1, j=1}^{M, M}|H_{ij}(0) - H_{ij}(t)|] \leq \frac{2M^2\sqrt{2}C}{\sqrt{\pi}} \tag{33}$$

Therefore, by Markov's inequality, given the probability $1 - \delta$, we get:

$$\sum_{i=1,j=1}^{M,M} |H_{ij}(0) - H_{ij}(t)| \leq \frac{2M^2\sqrt{2}C}{\sqrt{\pi}\delta} \tag{34}$$

In Du et al. (2019b), the authors prove that, given a small perturbation $K$:

$$\text{if } [\sum_{ij} |H_{ij}(0) - H_{ij}|] \leq K, \text{ then } \lambda_{min}(H) \geq \lambda_{min}(H(0)) - K \tag{35}$$

In our case, $K$ in Eq. 35 is given by $\frac{2M^2\sqrt{2}C}{\sqrt{\pi}\delta}$. Therefore,

$$\lambda_{min}(H(t)) \geq \lambda_{min}(H(0)) - \frac{2M^2\sqrt{2}C}{\sqrt{\pi}\delta} = \lambda_{min}(H(0)) - \frac{2\sqrt{2}M^2\eta t\sigma\sqrt{\Phi(1-\epsilon)}}{\sqrt{\pi}\delta} \tag{36}$$

We replace the term $\eta$ in Eq.36 with $\eta$'s upper bound given in the assumption of Theorem 3.2, i.e., $\eta < \frac{\lambda_0\sqrt{\pi}\delta}{2M^2\sqrt{2}\Phi(1-\epsilon)t\sigma}$, we can get that $\lambda_{min}(H(t))$ is always larger than 0; that is:

$$\lambda_{min}(H(t)) \geq \lambda_{min}(H(0)) - \frac{2\sqrt{2}M^2\eta t\sigma\sqrt{\Phi(1-\epsilon)}}{\sqrt{\pi}\delta} > 0 \tag{37}$$

This completes our proof. $\square$

## C   PROOF OF THEOREM 3.5

**Theorem 3.5** *Given a neural network with ReLU activation function optimized by minimizing Eq. 8, we assume that each initial weight vector $\{\boldsymbol{w_r}(0), r = 1, ..., n\}$ is i.i.d. generated from $\mathcal{N}(0, I)$ and the gradient for each weight follows an i.i.d. distribution $\mathcal{N}(0, \sigma)$. For some positive constants $\delta$ and $\epsilon$, if the learning rate $\eta$ satisfies $\eta < \frac{\lambda_0\sqrt{\pi}\delta}{2M^2\sqrt{2}\Phi(1-\epsilon)t\sigma}$, then with with probability at least $(1-\delta)(1-\epsilon)$, the following holds true: for any $r \in [m]$, $||\boldsymbol{w_r}(0) - \boldsymbol{w_r}(t)|| \leq C = \eta t\sigma\sqrt{\Phi(1-\epsilon)}$, and at training step $t$, the Gram matrix $H(t)$ satisfies:*

$$\lambda_{max}(H(t)) \leq \lambda_{max}(H(0)) + \frac{2\sqrt{2}M^2\eta t\sigma\sqrt{\Phi(1-\epsilon)}}{\sqrt{\pi}\delta} \tag{38}$$

$\Phi(\cdot)$ is the inverse cumulative distribution function for a $d$-degree chi-squared distribution $\chi^2(d)$.

The proof is similar to the proof of Theorem 3.2 (see Appendix B). We provide the entire proof below.

*Proof.* We first compute the probability of $||\boldsymbol{w_r}(0) - \boldsymbol{w_r}(t)|| \leq C$. Based on the assumption that $\{w_i(0), i = 1, ..., n\}$ follow i.i.d. $\mathcal{N}(0, I)$ and the gradient of each weight follows i.i.d. $\mathcal{N}(0, \sigma)$, considering the weight updating rule defined in Eq. 9 with learning rate $\eta$, each element in $\boldsymbol{w_r}(0) - \boldsymbol{w_r}(t)$ follows an i.i.d. $\mathcal{N}(0, \eta t\sigma)$. Therefore, $\frac{||\boldsymbol{w_r}(0)-\boldsymbol{w_r}||^2}{\eta^2 t^2\sigma^2}$ follows a chi-distribution with $d$ degrees of freedom $\chi^2(d)$:

$$\begin{aligned} P(||\boldsymbol{w_r}(0) - \boldsymbol{w_r}|| \leq C) &= P(||\boldsymbol{w_r}(0) - \boldsymbol{w_r}(t)||^2 \leq C^2) \\ &= P(\frac{||\boldsymbol{w_r}(0) - \boldsymbol{w_r}(t)||^2}{\eta^2 t^2\sigma^2} \leq \frac{C^2}{\eta^2 t^2\sigma^2}) \\ &= P(\frac{||\boldsymbol{w_r}(0) - \boldsymbol{w_r}(t)||^2}{\eta^2 t^2\sigma^2} \leq \Phi(1-\epsilon)) \\ &= 1 - \epsilon \end{aligned} \tag{39}$$

Given an input sample $\boldsymbol{x}_i$ and a weight vector $\boldsymbol{w_r}(t)$ from $\boldsymbol{W}(t)$, we define the following event:

$$\mathcal{A}_{ir} = \{||\boldsymbol{w_r}(t) - \boldsymbol{w_r}(0)|| \leq C\} \cap \{\mathbb{I}\{\boldsymbol{x_i^T}\boldsymbol{w_r}(0) \geq 0\} \neq \mathbb{I}\{\boldsymbol{x_i^T}\boldsymbol{w_r}(t) \geq 0\}\} \tag{40}$$

If $||\boldsymbol{w_r}(t) - \boldsymbol{w_r}(0)|| \leq C$ holds true, then:

$$
\begin{aligned}
\boldsymbol{x_i}^T \boldsymbol{w_r}(t) &= \boldsymbol{x_i}^T (\boldsymbol{w_r}(t) - \boldsymbol{w_r}(0)) + \boldsymbol{x_i}^T \boldsymbol{w_r}(0) \\
&= \text{sign}(\boldsymbol{x_i}^T(\boldsymbol{w_r}(t) - \boldsymbol{w_r}(0)))||\boldsymbol{w_r}(t) - \boldsymbol{w_r}(0)|| + \text{sign}(\boldsymbol{x_i}^T \boldsymbol{w_r}(0))||\boldsymbol{w_r}(0)||
\end{aligned}
\tag{41}
$$

Eq. 41 implies that if $||\boldsymbol{w_r}(0)||$ is larger than $||\boldsymbol{w_r}(t) - \boldsymbol{w_r}(0)||$, then $\boldsymbol{x_i}^T \boldsymbol{w_r}(0)$ determines the sign value of $\boldsymbol{x_i}^T \boldsymbol{w_r}(t)$. In other words, $\boldsymbol{x_i}^T \boldsymbol{w_r}(t)$ always has the same sign values as $\boldsymbol{x_i}^T \boldsymbol{w_r}(0)$; that is, $\mathbb{I}\{\boldsymbol{x_i}^T \boldsymbol{w_r}(0) \geq 0\} = \mathbb{I}\{\boldsymbol{x_i}^T \boldsymbol{w_r}(t) \geq 0\}$. Hence, if $||\boldsymbol{w_r}(t) - \boldsymbol{w_r}(0)|| \leq C$ and $\mathbb{I}\{\boldsymbol{x_i}^T \boldsymbol{w_r}(0) \geq 0\} \neq \mathbb{I}\{\boldsymbol{x_i}^T \boldsymbol{w_r}(t) \geq 0\}$ hold true, then $||\boldsymbol{w_r}(0)|| \leq C$. Therefore, the probability of event $\mathcal{A}_{ir}$:

$$
P(\mathcal{A}_{ir}) \leq P(\{||\boldsymbol{w_r}(0)|| \leq C\})
\tag{42}
$$

By the anti-concentration inequality of a Gaussian distribution Du et al. (2019b), we have:

$$
P(\mathcal{A}_{ir}) \leq P(\{||\boldsymbol{w_r}(0)|| \leq C\}) \leq \frac{\sqrt{2}C}{\sqrt{\pi}}
\tag{43}
$$

Therefore, if any weight vector $w_1, ..., w_m$ satisfies $||\boldsymbol{w_r}(0) - \boldsymbol{w_r}(t)|| \leq C$, we can bound the entry-wise deviation on the Gram matrix $H(t)$ at the training step $t$: for any $(i, j) \in [n] \times [n]$:

$$
\begin{aligned}
&\mathbb{E}[|H_{ij}(0) - H_{ij}(t)|] \\
=&\mathbb{E}[\frac{1}{m}|\boldsymbol{x_i}^T x_j \sum_{r=1}^{m} (\mathbb{I}\{\boldsymbol{x_i}^T \boldsymbol{w_r}(0) \geq 0, x_j^T \boldsymbol{w_r}(0) \geq 0\} - \mathbb{I}\{\boldsymbol{x_i}^T \boldsymbol{w_r}(t) \geq 0, x_j^T \boldsymbol{w_r}(t) \geq 0\})|] \\
=&\mathbb{E}[\frac{1}{m}|\boldsymbol{x_i}^T x_j \sum_{r=1}^{m} (\mathbb{I}\{\boldsymbol{x_i}^T \boldsymbol{w_r}(0) \geq 0\}\mathbb{I}\{x_j^T \boldsymbol{w_r}(0) \geq 0\} - \mathbb{I}\{\boldsymbol{x_i}^T \boldsymbol{w_r}(t) \geq 0\}\mathbb{I}\{x_j^T \boldsymbol{w_r}(t) \geq 0\})|]
\end{aligned}
\tag{44}
$$

We note that all the samples in the training set $\mathbb{S}$ (Eq. 1) are normalized with their L2-norm. Hence, we have both $||x_i|| = 1$ and $||x_j|| = 1$. Therefore, using the Cauchy–Schwarz inequality, the above equation is bounded as follows:

$$
\mathbb{E}[|H_{ij}(0) - H_{ij}(t)|] \leq \mathbb{E}[\frac{1}{m}\sum_{r=1}^{m}(\mathbb{I}\{\mathcal{A}_{ir} \cup \mathcal{A}_{jr}\}] \leq P(\mathcal{A}_{ir}) + P(\mathcal{A}_{jr})] \leq \frac{2\sqrt{2}C}{\sqrt{\pi}}
\tag{45}
$$

where the expectation is over the initial weight $w_r(0), r = \{1, ..., m\}$. Hence, considering all the elements in $H$, we have:

$$
\mathbb{E}[\sum_{i=1,j=1}^{M,M} |H_{ij}(0) - H_{ij}(t)|] \leq \frac{2M^2 \sqrt{2}C}{\sqrt{\pi}}
\tag{46}
$$

Therefore, by the Markov's inequality, given the probability $1 - \delta$, we get:

$$
\sum_{i=1,j=1}^{M,M} |H_{ij}(0) - H_{ij}(t)| \leq \frac{2M^2 \sqrt{2}C}{\sqrt{\pi}\delta}
\tag{47}
$$

Based on the matrix perturbation theory Bauer & Fike (1960); Eisenstat & Ipsen (1998), given a small perturbation $K$:

$$
\text{if } [\sum_{ij} |H_{ij}(0) - H_{ij}(t)|] \leq K, \text{ then } \lambda_{max}(H(t)) \leq \lambda_{max}(H(0)) + K
\tag{48}
$$

In our case, $K$ in Eq. 48 is given by $\frac{2M^2\sqrt{2}C}{\sqrt{\pi}\delta}$; that is:

$$
\lambda_{max}(H(t)) \leq \lambda_{max}(H(0)) + \frac{2\sqrt{2}M^2 \eta t \sigma \sqrt{\Phi(1-\epsilon)}}{\sqrt{\pi}\delta}
\tag{49}
$$

This completes our proof. □

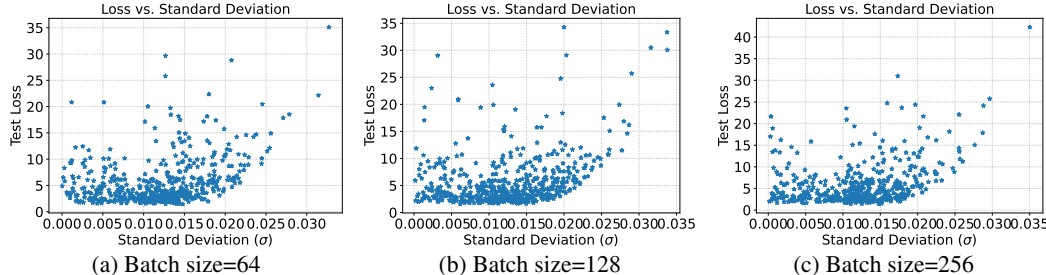

(a) Batch size=64          (b) Batch size=128          (c) Batch size=256

Figure 5: Test loss vs. standard deviation of gradients ($\sigma$ in Eq. 13) for randomly sampled 500 two-layer MLPs with ReLU on MNIST after one training epoch. We train these networks by minimizing the MSE loss between the output of networks and the real labels. As shown, the Networks with smaller $\sigma$ tend to have lower test loss values and thus have a better generalization capacity.

### C.1 SUPPLEMENTARY RESULTS: VALIDATION OF THEOREM 3.5

To empirically validate Theorem 3.5, we first create the training set $\mathbb{S}$ by normalizing the training samples in MNIST with their L2-norm. Next, we optimize a two-layer MLP with ReLU activation functions as defined in Eq. 7. We use the entire training set of MNIST and apply the gradient descent (Eq. 9) to update the weights. We vary the batch size as $\{64, 128, 256\}$ and measure the standard deviation of gradients ($\sigma$) w.r.t. parameters across different training batches. A very small learning rate of $\eta = 10^{-8}$ is set to satisfy the assumption in Theorem 3.5. Fig. 5 demonstrates the training loss after one epoch vs. standard deviation of gradients ($\sigma$). Clearly, the results show that if a network has a lower gradient standard deviation, then it tends to have lower test loss values, and thus, a better generalization capacity. These results empirically prove our claims in Theorem 3.5.

## D EXPERIMENTAL SETUP OF ZICO ON IMAGENET

### D.1 SEARCH SPACE

We use the commonly used MobileNetv2-based search space where the candidate networks are built by stacking multiple Inverted Bottleneck Blocks (IBNs) with SE modules Sandler et al. (2018); Pham et al. (2018); Lin et al. (2021); all the SE modules share the same se_ratio as 0.25. For each IBN, we vary the kernel size of the depth-wise convolutional layer from $\{3, 5, 7\}$ and sample the expansion ratio from $\{1, 2, 4, 6\}$. We consider ReLU as the activation function. For each point-wise convolutional layer, the range of the number of channels is from 8 to 1024 with a step size of 8. We use standard Kaiming_Init to initialize all linear and convolution layers for every candidate networks He et al. (2015).

### D.2 SEARCH ALGORITHM

We use an Evolutionary Algorithm (EA) to conduct the zero-shot NAS because it is concise and easy to implement[1] As shown in Algorithm 1, we search for the neural architectures with the highest ZiCo within the search space, given a specific budget $B$ (e.g., FLOPs). We repeat the search $T$ times; at each search step, we randomly select a structure from the candidate set $\mathbb{F}$ and mutate its architectures (e.g., kernel size, block type, number of blocks, and layer width) to generate a new network $F_i \in \mathcal{S}$. If the generated network $F_i$ meets the inference budget $B$, we calculate its ZiCo on $\mathbb{Z}$ and add $F_i$ to the candidate set $\mathbb{F}$. We remove the network with the smallest ZiCo from $\mathbb{F}$, if the number of architectures in $\mathbb{F}$ exceeds the threshold $E$. After $T$ steps, we select the network with the largest ZiCo as the final (optimal) architecture $F_P$.

---

[1]One can also use other methods to perform the search; see Appendix F.2.

---

**Algorithm 1** ZiCo-based zero-shot NAS framework

---

**INPUT:** Number of search steps $T$
        Inference budget $B$, Search space $\mathcal{S}$
        Set of input batch $\mathbb{Z} = \{(\boldsymbol{X_i}, \boldsymbol{y_i}), i = 1, 2\}$
        Population size $E$, Initial network $F_0 \in \mathcal{S}$
**OUTPUT:** Optimal network $F_P$
**SEARCH:**
Initialize $\mathbb{F} = \{F_0\}$
**for** $i = 1$ **to** $T$ **do**
    Randomly sample network $F_t$ from $\mathbb{F}$
    $F_i$ = randomly mutated architecture based on $F_t$ from $\mathcal{S}$
    **if** $F_i$ meets the inference budget $B$ **then**
        Compute ZiCo for $F_i$ on $\mathbb{Z}$ by Eq. 15
        Add $F_i$ to $\mathbb{F}$
        **if** $|\mathbb{F}| > E$ **then**
            Remove network with the smallest ZiCo from $\mathbb{F}$
        **end if**
    **end if**
**end for**
$F_P$= the network of the highest ZiCo in $\mathbb{F}$.

---

Specifically, we repeat the search $10^5$ times (i.e., $T = 10^5$) with the population size $E = 512$. For each of the candidate architectures, we compute ZiCo with two batches randomly sampled from the training set of ImageNet with batch size 128. In total, it takes 10 hours on a single NVIDIA 3090 GPU for $10^5$ search steps.

### D.3 TRAINING DETAILS

We use the same data augmentations configurations as in Pham et al. (2018): mix-up, label-smoothing, random erasing, random crop/resize/flip/lighting, and AutoAugment. We use the SGD optimizer with momentum 0.9 and weight decay 4e-5. We take EfficientNet-B3 as a teacher network and use the knowledge distillation method to train the network. We set the initial learning rate as 0.1 and used the cosine annealing scheme to adjust the learning rate during training. We train the obtained network 480 epochs, which takes 83 hours on a 40-core Intel Xeon CPU and 8 NVIDIA 3090 GPU-powered server.

## E SUPPLEMENTARY RESULTS ON NAS BENCHMARKS

### E.1 COMPARISON WITH MORE PROXIES

In this section, we further provide the comparison between our proposed ZiCo and more proxies proposed recently: KNAS (Xu et al. (2021)), NASWOT (Lopes et al. (2021)), GradSign ( Zhang & Jia (2022)), and NTK (TE-NAS Chen et al. (2021b), NASI Shu et al. (2022a)). To compute the correlations, we use the official code released by the authors of the above papers to obtain the values of these proxies[2]. As shown in Table 3, our proposed ZiCo performs better than all these proxies. For example, NASWOT and GradSign achieve a similar correlation score as ZiCo on NATSBench-TSS; however, ZiCo has a significantly higher correlation score than these two proxies on NATSBench-SSS.

Beside the correlation coefficients, we also report the optimal architectures found with various proxies. As shown in Table 4, the architectures found via ZiCo have the highest test accuracy on all these three datasets.

---

[2]NASI uses NTK to build their own search algorithms. Here, we directly compute the correlation between NTK and the real test accuracy.

Table 3: The correlation coefficients between various zero-cost proxies and two naive proxies (#Params and FLOPs) vs. test accuracy on NATSBench-SSS and NATSBench-TSS (KT and SPR represent Kendall's $\tau$ and Spearman's $\rho$, respectively). The results in italics represent the values of #Params' correlation coefficients. The results better than #Params are shown with bold fonts. Clearly, our proposed ZiCo is the only proxy that works consistently better than #Params and is generally the best among all these proxies. Both TE-NAS‡ (Chen et al. (2021b)) and NASI‡ (Chen et al. (2021b)) use NTK (Jacot et al. (2018)) as the accuracy proxy to build their own search algorithms.

| NATSBench-TSS (NASBench201) | | | | | | |
|---|---|---|---|---|---|---|
| **Dataset** | CIFAR10 | | CIFAR100 | | Img16-120 | |
| Proxy          Correlation | KT | SPR | KT | SPR | KT | SPR |
| Grad_norm Abdelfattah et al. (2021) | 0.46 | 0.63 | 0.47 | 0.63 | 0.43 | 0.58 |
| SNIP Lee et al. (2019b) | 0.46 | 0.63 | 0.46 | 0.63 | 0.43 | 0.58 |
| GraSP Wang et al. (2020) | 0.37 | 0.54 | 0.36 | 0.51 | 0.40 | 0.56 |
| Fisher Liu et al. (2021) | 0.40 | 0.55 | 0.41 | 0.55 | 0.37 | 0.50 |
| Synflow Tanaka et al. (2020) | 0.54 | 0.73 | 0.57 | 0.76 | 0.56 | 0.75 |
| KNAS Xu et al. (2021) | 0.14 | 0.20 | 0.24 | 0.35 | 0.30 | 0.42 |
| NASWOT Mellor et al. (2021) | **0.58** | **0.77** | **0.62** | **0.80** | **0.60** | **0.78** |
| NTK [TE-NAS Chen et al. (2021b), NASI Shu et al. (2022a)]‡ | 0.33 | 0.44 | 0.33 | 0.43 | 0.46 | 0.63 |
| GradSign Zhang & Jia (2022) | **0.58** | **0.77** | **0.59** | **0.79** | **0.59** | **0.78** |
| Zen-score Lin et al. (2021) | 0.29 | 0.38 | 0.28 | 0.36 | 0.29 | 0.40 |
| FLOPs | 0.54 | 0.73 | 0.51 | 0.71 | 0.49 | 0.67 |
| *#Params* | *0.57* | *0.75* | *0.55* | *0.73* | *0.52* | *0.69* |
| **ZiCo** | **0.61** | **0.80** | **0.61** | **0.81** | **0.60** | **0.79** |
| **NATSBench-SSS** | | | | | | |
| **Dataset** | CIFAR10 | | CIFAR100 | | Img16-120 | |
| Proxy          Correlation | KT | SPR | KT | SPR | KT | SPR |
| Grad_norm Abdelfattah et al. (2021) | 0.35 | 0.51 | 0.34 | 0.49 | 0.49 | 0.67 |
| SNIP Lee et al. (2019b) | 0.42 | 0.59 | 0.46 | 0.62 | 0.57 | 0.76 |
| GraSP Wang et al. (2020) | -0.09 | -0.13 | 0.01 | 0.01 | 0.29 | 0.42 |
| Fisher Liu et al. (2021) | 0.30 | 0.44 | 0.41 | 0.55 | 0.33 | 0.47 |
| Synflow Tanaka et al. (2020) | **0.61** | **0.81** | **0.60** | **0.80** | 0.39 | 0.57 |
| KNAS Xu et al. (2021) | 0.25 | 0.37 | 0.12 | 0.18 | 0.32 | 0.46 |
| NASWOT Mellor et al. (2021) | 0.45 | 0.63 | 0.43 | 0.59 | 0.42 | 0.59 |
| NTK [TE-NAS Chen et al. (2021b), NASI Shu et al. (2022a)]‡ | 0.17 | 0.26 | 0.04 | 0.06 | 0.20 | 0.30 |
| GradSign Zhang & Jia (2022) | 0.21 | 0.30 | 0.16 | 0.27 | 0.04 | 0.05 |
| Zen-score Lin et al. (2021) | 0.50 | 0.69 | 0.52 | 0.71 | **0.69** | **0.87** |
| FLOPs | 0.19 | 0.28 | 0.21 | 0.30 | 0.38 | 0.53 |
| *#Params* | *0.53* | *0.72* | *0.54* | *0.73* | *0.65* | *0.84* |
| **ZiCo** | **0.54** | **0.73** | **0.55** | **0.75** | **0.70** | **0.88** |

Table 4: The test accuracy of optimal architectures obtained by various zero-shot proxies (averaged over 5 runs) on NATSBench-TSS search space. The best results are shown with bold fonts.

| CIFAR100 | Groud Truth | Grad_norm | SNIP | GraSP | Fisher | Jacob_cov | Synflow | Zen-score | #Params | FLOPs | **ZiCo** |
|---|---|---|---|---|---|---|---|---|---|---|---|
| | *73.5* | 60.0 | 60.0 | 60.0 | 60.0 | 68.9 | 71.1 | 68.1 | **71.1** | 71.1 | **71.1±0.3** |
| Img16-120 | Groud Truth | Grad_norm | SNIP | GraSP | Fisher | Jacob_cov | Synflow | Zen-score | #Params | FLOPs | **ZiCo** |
| | *47.3* | 29.3 | 29.3 | 5.5 | 29.3 | 25.1 | 41.2 | 40.8 | 41.4 | 41.4 | **41.8±0.3** |
| CIFAR10 | Groud Truth | Grad_norm | SNIP | GraSP | Fisher | Jacob_cov | Synflow | Zen-score | #Params | FLOPs | **ZiCo** |
| | *94.5* | 89.5 | 89.5 | 89.5 | 89.5 | 88.4 | 90.4 | 90.6 | 93.7 | 93.7 | **94.0±0.4** |

## E.2 COMPARISON ON TRANSNAS-BENCH-101-MICRO

In this section, we compare our proposed ZiCo against existing proxies on more diverse tasks. We compare our proposed ZiCo against existing proxies on one mainstream NAS benchmark *TransNAS-Bench-101* Duan et al. (2021). We pick the largest search space TransNAS-Bench-101-Micro which contains 4096 total architectures with different cell structures. We compare ZiCo with various proxies under the following four tasks:

- **Scene Classification.** Scene classification is a 47-class classification task that predicts the room type in the image.

Table 5: The correlation coefficients under different proxies vs. test performance on TransNAS-Bench-101-Mirco. Clearly, our proposed ZiCo is consistently very close to the best score (only 0.01 or 0.02 lower score) except for Autoencoding (still, ZiCo is the second best on Autoencoding). Though Fisher works better than ZiCo on Autoencoding, ZiCo has a significantly higher score on the rest of tasks. We note that existing proxies do not achieve a high correlation on all tasks consistently.

| | Autoencoding | | Scene Classification | |
|---|---|---|---|---|
| Proxy | Kendall's $\tau$ | Spearman's $\rho$ | Kendall's $\tau$ | Spearman's $\rho$ |
| Grad_norm Abdelfattah et al. (2021) | 0.24 | 0.32 | 0.47 | 0.65 |
| SNIP Lee et al. (2019b) | 0.20 | 0.27 | 0.52 | 0.71 |
| Grasp Wang et al. (2020) | 0.09 | 0.14 | 0.19 | 0.28 |
| Fisher Liu et al. (2021) | **0.42** | **0.59** | 0.49 | 0.67 |
| Synflow Tanaka et al. (2020) | 0.00 | 0.00 | **0.53** | **0.72** |
| NASWOT Lopes et al. (2021) | 0.01 | 0.02 | 0.43 | 0.60 |
| Zen-score Lin et al. (2021) | 0.09 | 0.14 | 0.52 | 0.72 |
| GradSign Zhang & Jia (2022) | 0.01 | 0.02 | 0.32 | 0.46 |
| Params | 0.01 | 0.01 | 0.46 | 0.64 |
| FLOPs | 0.02 | 0.02 | 0.47 | 0.65 |
| **ZiCo** (Ours) | **0.24** | **0.35** | **0.51** | **0.71** |

| | Jigsaw | | Surface Normal | |
|---|---|---|---|---|
| Proxy | Kendall's $\tau$ | Spearman's $\rho$ | Kendall's $\tau$ | Spearman's $\rho$ |
| Grad_norm Abdelfattah et al. (2021) | 0.23 | 0.35 | 0.24 | 0.36 |
| SNIP Lee et al. (2019b) | 0.27 | 0.41 | 0.32 | 0.49 |
| Grasp Wang et al. (2020) | 0.07 | 0.11 | 0.01 | 0.01 |
| Fisher Liu et al. (2021) | 0.19 | 0.30 | 0.10 | 0.14 |
| Synflow Wang et al. (2020) | 0.32 | 0.47 | 0.00 | 0.00 |
| NASWOT Lopes et al. (2021) | 0.29 | 0.42 | 0.41 | 0.57 |
| Zen-score Lin et al. (2021) | 0.35 | 0.50 | **0.52** | **0.71** |
| GradSign Zhang & Jia (2022) | **0.38** | **0.53** | 0.29 | 0.40 |
| Params | 0.29 | 0.44 | 0.45 | 0.63 |
| FLOPs | 0.30 | 0.45 | 0.46 | 0.64 |
| **ZiCo** (Ours) | **0.36** | **0.52** | **0.50** | **0.68** |

- **Jigsaw.** In the Jigsaw task, the input image is divided into nine patches and shuffled based on one of 1,000 predefined permutations. The target here is to classify which permutation is used.

- **Autoencoding.** Autoencoding is a pixel-level prediction task that encodes an input image into a low-dimension embedding vector and then reconstructs the raw image from the vector.

- **Surface Normal.** Similar to autoencoding, surface normal is a pixel-level prediction task that predicts surface normal statistics.

As shown in Table 5, ZiCo consistently works well on Scene Classification, Jigsaw, and Surface Normal; ZiCo has only 0.01 or 0.02 lower correlation scores than the highest scores. Though Fisher works better than ZiCo on Autoencoding, ZiCo has significantly higher correlation scores than Fisher on the remaining three tasks. One possibility why Fisher works best on Autoencoding is that Autoencoding is an image-to-image task; Fisher is the only proxy that is built on the gradient w.r.t. feature maps and thus can better extract the information between the input and output images. Although Fisher works better than ZiCo on Autoencoding (we are still second best), ZiCo has a significantly higher score on the remaining tasks. As shown in the main paper, we again note that existing proxies do not achieve a high correlation on all tasks consistently.

Table 6 demonstrates the test accuracy of the best architectures found using various proxies on each of the above tasks in TransNAS-Bench-101-Micro. Once again, we see that ZiCo significantly outperforms existing proxies on all tasks except Autoencoding, where we trail Fisher by only 0.01 SSIM. Nonetheless, ZiCo is second best on the Autoencoding task. Note that, similar to the correlation results in Table 5, other proxies do not consistently achieve high accuracy. For instance, while methods like Synflow or Zenscore achieve results close to ours on Scene Classification and Surface

Table 6: The test performance of optimal architectures obtained by various zero-shot proxies (averaged over 5 runs) on TransNAS-Bench-101-Micro search space. The best results are shown with bold fonts.

| | Autoencoding | Scene Classification | Jigsaw | Surface Normal |
|---|---|---|---|---|
| Metric | SSIM | Accuracy | Accuracy | SSIM |
| Ground Truth | 0.58 | 54.9 | 95.4 | 0.59 |
| Grad_norm | 0.36± 0.03 | 48.7±0.7 | 80.3±0.3 | 0.53±0.00 |
| SNIP | 0.33±0.04 | 48.7±1.1 | 80.3±0.1 | 0.53±0.01 |
| Grasp | 0.33±0.06 | 50.2±1.6 | 91.1±0.3 | 0.38±0.06 |
| Fisher | **0.49±0.01** | 48.7±0.6 | 83.5±1.2 | 0.31±0.03 |
| Synflow | 0.46±0.07 | **53.7±1.2** | 90.9±0.4 | **0.57±0.06** |
| NASWOT | 0.43±0.02 | 53.2±0.6 | 92.3±0.3 | 0.53±0.02 |
| Zen-score | 0.46±0.01 | **53.7±0.2** | 87.5±0.4 | 0.55±0.00 |
| GradSign | 0.35±0.03 | 53.6±0.4 | 93.1±0.4 | **0.57±0.02** |
| Params | 0.46 | 53.70 | 85.90 | 0.55 |
| FLOPs | 0.46 | 53.70 | 85.90 | 0.55 |
| **ZiCo (Ours)** | 0.48±0.02 | **53.7±0.4** | **93.2±0.4** | **0.57±0.01** |

Table 7: The correlation coefficients under three different proxies vs. test accuracy on NATSBench-SSS (KT and SPR represent Kendall's $\tau$ and Spearman's $\rho$, respectively). Clearly, our proposed ZiCo works consistently better than using mean only and STD only on all these datasets.

| Dataset | CIFAR10 | | CIFAR100 | | Img16-120 | |
|---|---|---|---|---|---|---|
| Method | KT | SPR | KT | SPR | KT | SPR |
| Mean Only | 0.25 | 0.37 | 0.39 | 0.55 | 0.61 | 0.81 |
| STD only | 0.39 | 0.55 | 0.42 | 0.6 | 0.45 | 0.62 |
| **ZiCo (Mean + STD)** | **0.54** | **0.73** | **0.55** | **0.75** | **0.70** | **0.88** |

Normal, they produce poor results on other tasks like Jigsaw. Therefore, ZiCo consistently performs well on highly different tasks.

### E.3 ILLUSTRATION OF VARIOUS PROXIES VS. REAL TEST ACCURACY

We provide some illustration figures of real test accuracy vs. various proxies on NATSBench-SSS search space for CIFAR10 (Fig. 6) and ImageNet16-120 datasets(Fig. 7). We also show the same illustrative results (real test accuracy vs. various proxies) on NASBench101 search space in Fig. 8.

## F ABLATION STUDY

### F.1 IMPACT OF MEAN AND STD

We randomly select 2000 networks from NATSBench-SSS on CIFAR10, CIFAR100, and Img16-120 datasets and compute the following proxies: (i) Mean value of gradients only; (ii) Standard deviation (STD) value of gradients only; (iii) Combination of mean and std value, i.e., our proposed *ZiCo*. We then calculate the correlation coefficients between these proxies and the real test accuracy. As shown in Table. 7, our proposed ZiCo performs better on these three datasets than either using mean only or STD only. Therefore, our proposed ZiCo is a better-designed proxy than using mean or STD individually.

### F.2 SEARCH ALGORITHMS: ZERO-COST PT

In this section, we demonstrate that our proposed ZiCo can be combined with other search algorithms. We take the Zero-Cost-PT (Zero-PT) as an example Xiang et al. (2021b) because it is specifically designed for zero-shot proxies and is very time-efficient. Essentially, Zero-PT first integrates all candidate networks into a supernet and assigns learnable weights to each candidate operation (same as one-shot NAS). Then Zero-PT uses the zero-cost proxy instead of the training

Table 8: The test accuracy of optimal architectures obtained by various zero-shot proxies (average on 5 runs) on NATSBench-TSS search space. The best results are shown with bold fonts.

| Proxy | CIFAR10 | CIFAR100 | Img16-120 | Costs(GPU hours) |
|---|---|---|---|---|
| Zero-PT+SNIP Lee et al. (2019b) | 93.52±0.18 | 70.75±0.19 | 44.45±0.14 | 0.10 |
| Zero-PT+NASWOT Lopes et al. (2021) | 93.42±0.07 | 70.77±0.51 | 45.11±0.26 | 0.11 |
| Zero-PT+Synflow Tanaka et al. (2020) | 87.68±0.16 | 58.92±0.17 | 32.20±0.00 | 0.13 |
| Zero-PT+KNAS Xu et al. (2021) | 93.95±0.03 | 72.44±0.26 | 46.01±0.12 | 0.10 |
| Zero-PT+Grad_norm Abdelfattah et al. (2021) | 93.52±0.18 | 70.75±0.30 | 44.48±0.11 | 0.07 |
| Zero-PT+Zen-score Lin et al. (2021) | 93.84±0.05 | 71.63±0.06 | **46.67±0.16** | **0.02** |
| Zero-PT+GradSign Zhang & Jia (2022) | 93.76±0.12 | 71.11±0.23 | 42.95±1.29 | 0.06 |
| **Zero-PT+ZiCo (Ours)** | **94.15±0.22** | **72.77±0.66** | 46.39±0.23 | 0.12 |

Table 9: Comparison of Top-1 accuracy of our ZiCo-based NAS against NAS methods with standalone training on ImageNet under various FLOP budgets. For the 'Method' column, 'MS' represents multi-shot NAS; 'OS' is short for one-shot NAS; Scaling represents network scaling methods; 'ZS' is short for zero-shot NAS. 'no KD' means we train the network without Knowledge Distillation (KD); '150E' means we train the network with 150 epochs, similar for 350E. The results are averaged over three suns. We note that some NAS methods use knowledge distillation to improve the test accuracy; hence, we remove those methods from this table. The results are averaged over three runs.

| Budget (maximal #FLOPs) | Approach | FLOPs | Top-1 | Method | Costs[GPU Days] |
|---|---|---|---|---|---|
| | EfficientNet-B0 Tan & Le (2019) [350E] | 390M | 77.1 | Scaling | 3800 |
| | EfficientNet-B0 Tan et al. (2019)[150E] | 390M | 76.0 | Scaling | 3800 |
| | MnasNet-A3 Tan et al. (2019) | 403M | 76.7 | MS | - |
| | BN-NAS Chen et al. (2021a) | 470M | 75.7 | MS | 0.8 |
| 450M | RLNAS Zhang et al. (2021) | 473M | 75.6 | OS | - |
| | NASNet-B Zoph et al. (2018) | 488M | 72.8 | MS | 1800 |
| | CARS-D Yang et al. (2020) | 496M | 73.3 | MS | 0.4 |
| | Zen-score Lin et al. (2021) [no KD; 150E] | 410M | 75.6 | ZS | 0.5 |
| | #Params | 451M | 63.5 | ZS | 0.02 |
| | **ZiCo (Ours) [no KD; 150E]** | **448M** | **76.5±0.2** | **ZS** | **0.4** |
| | DARTS Liu et al. (2019) | 574M | 73.3 | OS | 4 |
| | NAO Luo et al. (2018) | 584M | 75.5 | MS | 58.3 |
| | PC-DARTS Xu et al. (2019) | 586M | 75.8 | OS | 3.8 |
| | PNAS Liu et al. (2018a) | 588M | 74.2 | MS | 224 |
| | CARS-I Yang et al. (2020) | 591M | 75.2 | MS | 0.4 |
| | EnTranNAS Yang et al. (2021) | 594M | 76.2 | OS | 2.1 |
| | ProxylessNAS Cai et al. (2019) | 595M | 76.0 | OS | 8.3 |
| 600M | RLNAS Zhang et al. (2021) | 597M | 75.9 | OS | - |
| | MAGIC-AT Xu et al. (2022) | 598M | 76.8 | OS | 2 |
| | SemiNAS Luo et al. (2020) | 599M | 76.5 | MS | 4 |
| | EfficientNet-B1 Tan et al. (2019)[350E] | 700M | 79.1 | Scaling | 3800 |
| | EfficientNet-B1 Tan et al. (2019)[150E] | 700M | 77.4 | Scaling | 3800 |
| | TE-NAS Chen et al. (2021b) | 599M | 75.5 | ZS | 0.17 |
| | Zen-score Lin et al. (2021) [no KD; 150E] | 611M | 76.1 | ZS | 0.5 |
| | **ZiCo (Ours) [no KD; 150E]** | **603M** | **77.1±0.3** | **ZS** | **0.4** |

accuracy to update the weights for each candidate operation. The final architecture is generated by selecting the operations with the highest weight values.

We combine different accuracy proxies with Zero-PT under the NASBench-201 and report the optimal architectures found with various proxies[3]. As shown in Table 8, the architectures found via ZiCo have the highest test accuracy except for Img16-120 datasets (ZiCo is the second best on Img16-120)).

### F.3 TRAINING RECIPE: WITHOUT DISTILLATION

In this section, we train the obtained network under various FLOPs budgets with the exact same training setup as Xu et al. (2022); Cai et al. (2019). Specifically, we train the neural network for 150 epochs with batch size 512 and input resolution 224×224. We train the network without knowledge

---

[3]We implement the code ourselves since the authors have not released the code yet. The difference between Table 4 and Table 8 comes from the search algorithm: Table 4 uses traversal search among all candidate networks; Table 8 uses perturbation-based zero-cost PT Xiang et al. (2021b).

Table 10: Comparison of Top-1 accuracy of our ZiCo-based NAS against NAS methods with standalone training on CIFAR10 on DARTS search space. For the 'Method' column,'MS' represents multi-shot NAS; 'OS' is short for one-shot NAS; 'ZS' is short for zero-shot NAS. '600E' means we train the network with 600 epochs, similar to 800E. The results are averaged over three suns. The results are averaged over three runs.

| Approach | Test Error (%) | Method | Cost(GPU days) |
|---|---|---|---|
| AmoebaNet-A Real et al. (2019) | 3.34±0.06 | MS | 3150 |
| PNAS Liu et al. (2018a) | 3.41±0.09 | MS | 225 |
| ENAS Tan & Le (2019) | 2.89 | MS | 0.5 |
| NASNet-A Zoph et al. (2018) | 2.65 | MS | 2000 |
| DARTS-v1 Liu et al. (2019) | 3.00±0.14 F | OS | 0.4 |
| DARTS-v2 Liu et al. (2019) | 2.76±0.09 | OS | 1 |
| SNAS Xie et al. (2019) | 2.85±0.02 | OS | 1.5 |
| GDAS Dong & Yang (2019) | 2.82 | OS | 0.17 |
| BayesNAS Zhou et al. (2019) | 2.81±0.04 | OS | 0.2 |
| ProxylessNAS Cai et al. (2019) | 2.08 | OS | 4 |
| P-DARTS Chen et al. (2019) | 2.5 | OS | 0.3 |
| PC-DARTS Xu et al. (2019) | 2.57±0.07 | OS | 0.1 |
| SDARTS-ADV Chen & Hsieh (2020) | 2.61±0.02 | OS | 1.3 |
| Zen-score Lin et al. (2021) | 2.55±0.04 | ZS | 0.01 |
| TE-NAS Chen et al. (2021b) | 2.63±0.064 | ZS | 0.05 |
| ZiCo(ours) | 2.45±0.11 | ZS | 0.03 |

distillation and do *not* use advanced data augmentation methods (e.g., mixup, RandAugment, etc). Finally, we set the initial learning rate as 0.4 with a cosine annealing scheduling scheme. Moreover, we train EfficientNets and the previous SOTA zero-shot NAS approach (Zen-score) under the same setup.

As shown in Table 9, ZiCo outperforms all of the previous zero-shot NAS approaches. For example, when the FLOPs budget is around 600M, ZiCo achieves 77.1% Top-1 accuracy, which is 1.0% and 1.6% higher than previous SOTA zero-shot NAS methods, i.e., Zen-score, and TE-NAS, respectively. Moreover, ZiCo finds a model with similar accuracy as EfficientNet-B1, but with 100M fewer FLOPs and much less search cost. Overall, compared to the regular one-shot or multi-shot NAS methods, ZiCo achieves comparable or higher test accuracy with 5-9500× less search time.

## F.4 SEARCH SPACE: DARTS

In this section, we use ZiCo to conduct the zero-shot NAS on the DARTS search space. We first use Algorithm 1 to find the networks with the highest ZiCo without FLOPs budgets on the CIFAR10 dataset. We conduct the search for 100k steps; this takes 0.7 hours on a single NVIDIA 3090 GPU (i.e., 0.03 GPU days). Then, we train the obtained network with the exact same training setup as the original DARTS paper Liu et al. (2019)[4]; specifically, we train the neural network for 600 epochs with a batch size of 128. We only use the standard data augmentation (normalization, cropping, and random flipping) together with the cutout tricks. We don't use knowledge distillation or any other advanced data augmentation tricks. Finally, we set the initial learning rate as 0.025 with a cosine annealing scheduling scheme. We repeat the same experiments for Zen-score.

As shown in Table 10, ZiCo outperforms previous zero-shot NAS approaches, e.g, Zen-score and TE-NAS. Moreover, compared to the regular one-shot or multi-shot NAS methods, ZiCo achieves comparable or higher test accuracy with at least 10× less search time.

---

[4]Most of the baseline approaches in Table 10 use the same setup as ours.

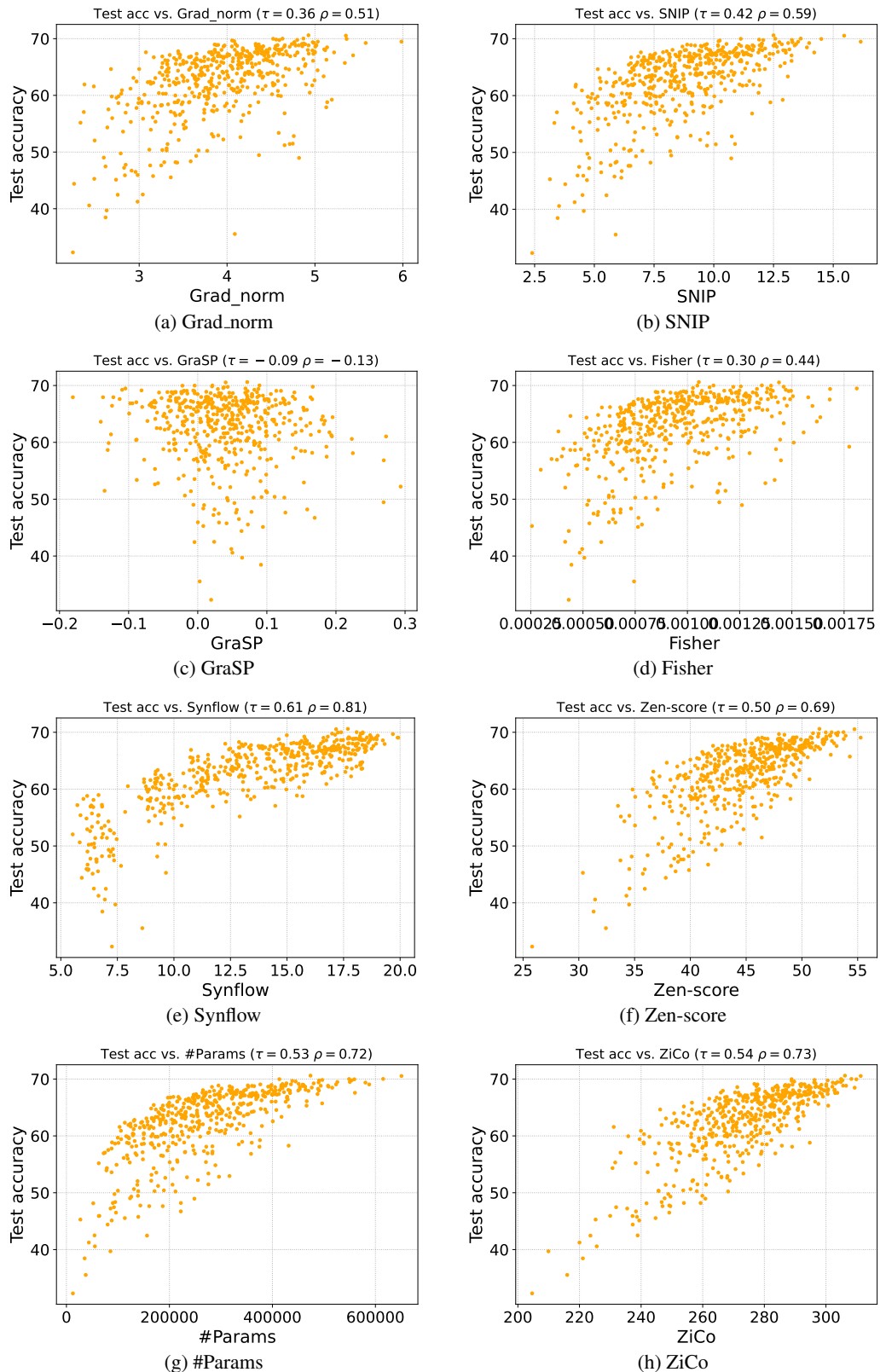

Figure 6: Real test accuracy vs. various proxies on NATSBench-SSS search space for CIFAR10 dataset. $\tau$ and $\rho$ are short for Kendall's $\tau$ and Spearman's $\rho$, respectively.

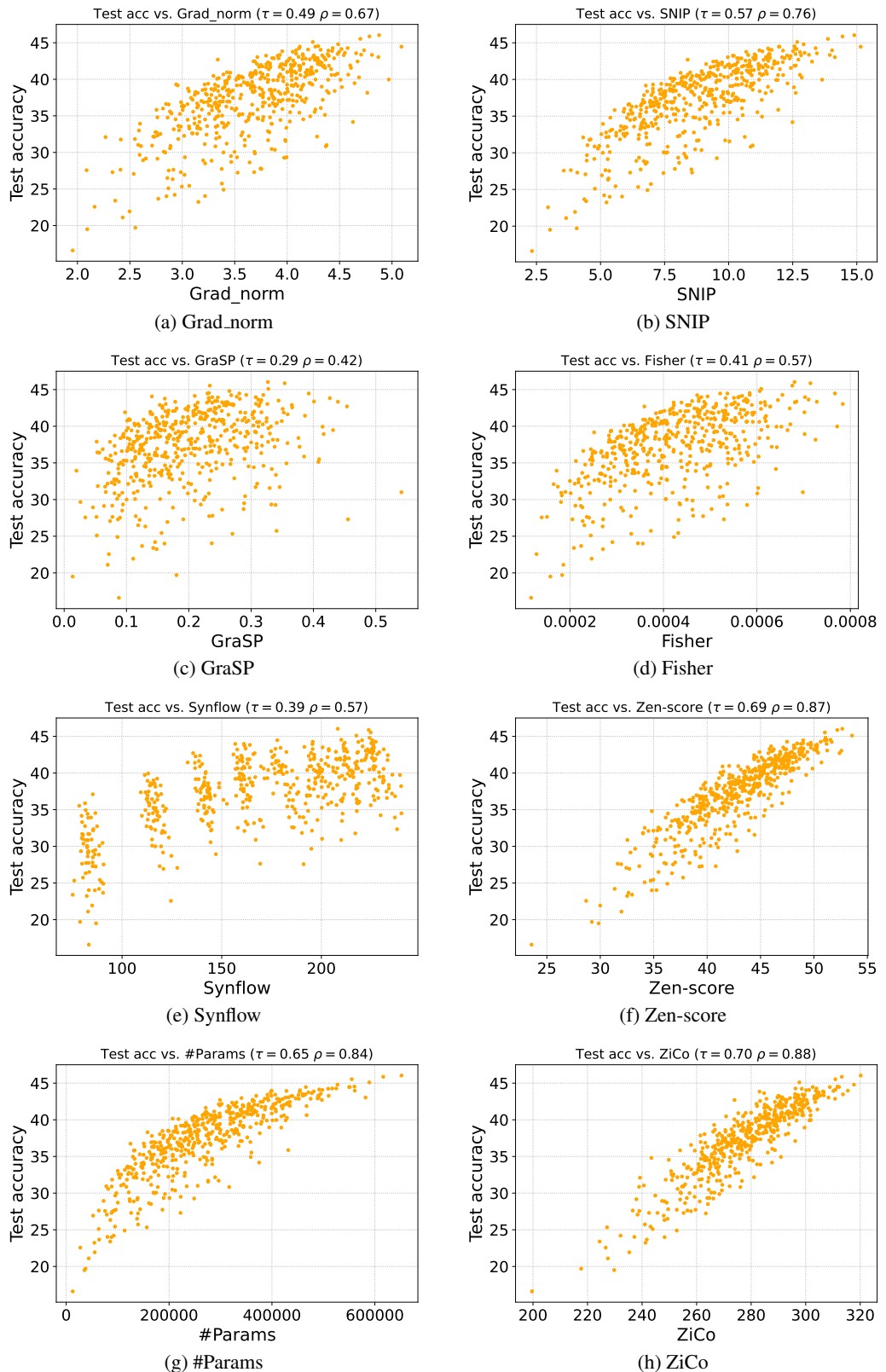

Figure 7: Real test accuracy vs. various proxies on NATSBench-SSS search space for ImageNet16-120 dataset. $\tau$ and $\rho$ are short for Kendall's $\tau$ and Spearman's $\rho$, respectively.

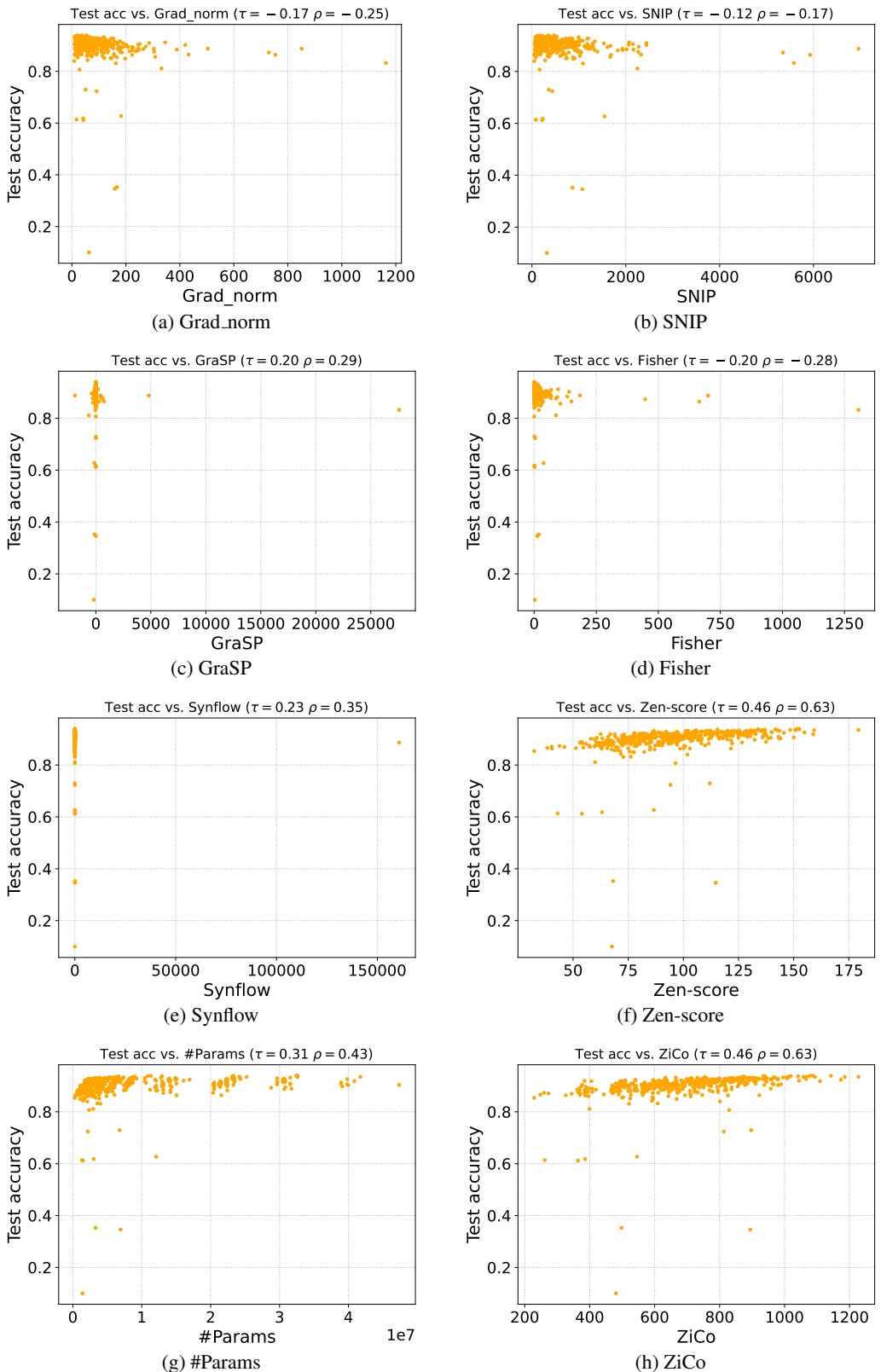

Figure 8: Real test accuracy vs. various proxies on NASBench101 search space for CIFAR10 dataset. $\tau$ and $\rho$ are short for Kendall's $\tau$ and Spearman's $\rho$, respectively.

