# OpenReview forum: "ZiCo: Zero-shot NAS via inverse Coefficient of Variation on Gradients"
_ICLR.cc/2023/Conference — ICLR 2023 notable top 25%_

### Official Review · Reviewer_AcGU · 2022-10-23

**Confidence:** 3
**Correctness:** 4
**Technical Novelty And Significance:** 2
**Empirical Novelty And Significance:** 3
**Recommendation:** 6

**Clarity, Quality, Novelty And Reproducibility:**

Clarity and Quality:
The paper is well-written and easy to follow.

Novelty:
While the theoretical findings are not surprising, the resulting metric and algorithm are intuitively sound and empirically strong and consistent.

Reproducibility:
The source code is provided in the supp material.

**Strength And Weaknesses:**

Strength:

- The proposed method is theoretically inspired. The author provides an extensive principled analysis of how gradient statistics contribute to the convergence of the network.
- The author provides extensive evaluations of the proposed metrics across popular benchmarks and also on MobileNet space. Empirical results are strong and consistent.

Weakness:

- [minor] Though extensive analysis, the theoretical findings are not too surprising: i.e. stabler and non-vanishing gradients help with network training.
- While the paper presents detailed theoretical results on how gradient statistics affect convergence, it might not necessarily lead to better generalization. It is often the case that there are networks that converge fast but plateau quickly [1]. I’d like to hear the author’s take on potential ways to connect the current results with generalization errors.

Question:

- For ImageNet results, it is sometimes hard to make sense of the improvement since different methods could be using drastically different search spaces and training protocols. I wonder which specific training protocol the author uses to achieve the reported accuracy. Also, whether the search space is identical to some previous works or modified?

[1] Zhou et al. Theory-Inspired Path-Regularized Differential Network Architecture Search. NeurIPS 2020

**Summary Of The Paper:**

This paper studies train-free metrics for NAS.

The performance of existing train-free metrics is often inconsistent across different search spaces, less consistent than even simply counting parameters.

This paper presents a new train-free metric based on a theoretical analysis on the correlation of gradient statistics to network convergence.

Extensive empirical results are provided on a wide range of popular benchmarks and also importantly on MobileNet space and ImageNet.

While the results do not always outperform existing train-free metrics, it is much more stable across different search spaces, which is far more important.

**Summary Of The Review:**

The paper is solving an important and relevant problem of train-free NAS metrics: their consistency issue across different search spaces.
The proposed method is simple yet theoretically grounded. I vote for acceptance.

---

> ### Author Response · Authors · 2022-11-13
> **Response to Reviewer AcGU**
>
> **Q1 While the paper presents detailed theoretical results on how gradient statistics affect convergence, it might not necessarily lead to better generalization. It is often the case that there are networks that converge fast but plateau quickly. I’d like to hear the author’s take on potential ways to connect the current results with generalization errors.**
>
> We provide a new theorem (Theorem C.1, Page 21) that links the standard deviations of gradients with the generalization capacity of neural networks. In Theorem C.1, we prove that the networks with lower standard deviations of gradients have a better generalization capacity, and thus a better test performance.  Hence, our proposed proxy can indicate both the convergence rate and generalization capacity of neural networks. We provide the details of the proof and experimental results in Appendix C (Page 21).
>
> **Q2 For ImageNet results, it is sometimes hard to make sense of the improvement since different methods could be using drastically different search spaces and training protocols. I wonder which specific training protocol the author uses to achieve the reported accuracy. Also, whether the search space is identical to some previous works or modified?**
>
> The training protocol for results in Table 3 (Page 8) is exactly the same as in previous work [1,2]. We provide more results under different training protocols in Appendix F.3 (Page 28). The search space used in our paper is a commonly used MobileNet-v2-like search space with inverted bottleneck residual blocks and Squeeze-and-Excitation modules, which is the same as previous work [1,2,3,4,5,6].
>
>
> **Q3 [minor] Though extensive analysis, the theoretical findings are not too surprising: i.e. stabler and non-vanishing gradients help with network training.**
>
> We provide a new theorem (Theorem C.1, Page 21) that links the standard deviations of gradients with the generalization capacity of neural networks. Please check our response to the Q1. We believe the new theoretical insight between the gradient analysis and generalization capacity of neural networks is a major contribution our paper brings to this area.
>
> We hope that our response fully addresses Reviewer’s questions so Reviewer would consider raising the ratings for our work.
>
> [1] Pham, Hieu, et al. "Efficient neural architecture search via parameters sharing." ICML 2018.
>
> [2] Lin, Ming, et al. "Zen-nas: A zero-shot nas for high-performance image recognition." ICCV 2021.
>
> [3] Cai, Han, et al. "Once-for-all: Train one network and specialize it for efficient deployment." ICLR 2020.
>
> [4] Yu, Jiahui, et al. "Bignas: Scaling up neural architecture search with big single-stage models." ECCV 2020.
>
> [5] Cai, Han, et al. "Proxylessnas: Direct neural architecture search on target task and hardware." ICLR 2019.
>
> [6] Tan, Mingxing, et al. "Efficientnet: Rethinking model scaling for convolutional neural networks." ICML 2019.

---

> > ### Author Response · Authors · 2022-11-16
> > **Sincerely expecting further discussions with reviewer AcGU**
> >
> > Dear Reviewer AcGU:
> >
> > We want to thank you for the constructive comments in your review. As a follow-up to our responses, we would like to kindly remind you that the discussion period is ending soon. We hope to use this open response period to discuss the paper to solve the concerns and improve the quality of our paper. Have you gotten a chance to read our responses above, which attempt to address all of your concerns?
> >
> > We would be more than happy to provide more information or clarification and hope that they could lead to a positive and fair assessment of this paper.
> >
> > Best,
> >
> > Authors of paper 2989

---

> > > ### Comment · Reviewer_AcGU · 2022-11-22
> > > **Reply to authors**
> > >
> > > I thank the authors for their response. I think the community can benefit from the method this paper presents, so I would like to keep my score as accept.

---

> > > > ### Author Response · Authors · 2022-11-23
> > > > **Thanks for the reply**
> > > >
> > > > Dear Reviewer AcGU,
> > > >
> > > > We are glad that your concerns were resolved and you want our paper to be accepted. According to your feedback, could you please let us know if there is something else we can improve in our response that may help you increase the rating for our work?
> > > >
> > > > Thanks again!
> > > >
> > > > Authors of paper 2989

---

### Official Review · Reviewer_9avK · 2022-10-24

**Confidence:** 4
**Correctness:** 3
**Technical Novelty And Significance:** 3
**Empirical Novelty And Significance:** 2
**Recommendation:** 6

**Clarity, Quality, Novelty And Reproducibility:**

In general, this paper is well-written and easy to follow. Meanwhile, the theoretical perspective is new for the NAS area.

**Strength And Weaknesses:**

Strengths:
1. Overall, this paper is well-written and well-motivated.
2. The proposed method in this paper is motivated by the theoretical convergence of linear models and DNNs.
3. Extensive results in this paper have validated the effectiveness and efficiency of the proposed method.

Weaknesses:
1. While this paper is motivated by the inconsistent performance of existing zero-shot proxies, the reason why the proposed method is able to tackle this problem has not been well explained either from the empirical perspective or the theoretical perspective.
2. There is a gap between the derived theoretical convergence and the proposed zero-shot proxy in eq.14. Firstly, Theorem 3.1 and 3.2 are the bounds for step t during the model training with SGD. This means that the convergences of linear models and DNNs are related to not only the initialization but also the model parameters during training and hence can not be simply determined by initialization according to the theorems in this paper. So, without further justification, simply using eq.14 alone may not characterize the convergence of DNNs well theoretically. Secondly, it seems that the mathematical form of eq.14 can not be directly derived from Theorem 3.1 and 3.2, i.e., the form of coefficient of variation doesn't appear anywhere in these theorems. So, the motivation for using the form of eq.14 is kind of weak to me. Thirdly, Theorem 3.1 and 3.2 can only characterize the convergence of models whereas we need to characterize the generalization performance of architectures in NAS. So, is there any other justification that authors can provide to support why characterizing convergence is already enough?
3. This paper misses the comparison with other training-free NAS works [1,2] in their experiments.
4. While this paper is motivated by the inconsistent performance of existing zero-shot proxies, I recommend more search results on other diverse tasks (e.g., the ones mentioned in (White et al., 2022)) to support the improved consistency achieved by ZiCo.

[1] GradSign: Model Performance Inference with Theoretical Insights
[2] NASI: Label- and Data-agnostic Neural Architecture Search at Initialization

**Summary Of The Paper:**

While existing training-free proxies can usually achieve compelling search results, they typically can not work consistently better than #Params in practice (White et al., 2022). So, this paper aims to improve such a state of affairs. Specifically, this paper firstly theoretically proves that the convergence of DNNs is highly correlated to the mean and variance of the gradient at initialization with respect to different input samples. Inspired by such a finding, this paper then proposes to use the inverse coefficient of variation on gradients as the zero-shot proxy for training-free NAS. Empirical results show that the proposed method in this paper can improve over other zero-shot proxies.

**Summary Of The Review:**

Overall, my major concerns lie in the theoretical part and the empirical comparison of this paper. I hope the authors can address my concerns during the rebuttal period.

---

> ### Author Response · Authors · 2022-11-13
> **Response to Reviewer 9avK (1/2)**
>
> **Q1 There is a gap between the derived theoretical convergence and the proposed zero-shot proxy in eq.14. Theorem 3.1 and 3.2 are the bounds for step t during the model training with SGD. This means that the convergences of linear models and DNNs are related to not only the initialization but also the model parameters during training and hence can not be simply determined by initialization according to the theorems in this paper. So, without further justification, simply using eq.14 alone may not characterize the convergence of DNNs well theoretically.**
>
> We understand the reviewer’s concern. Indeed, analyzing the gradient changes between different training steps is ideal for understanding the convergence property of networks. However, there is an important consideration: in zero-shot NAS, training is strictly forbidden (this is why it’s called zero-shot after all). To tackle this challenge, we approximate the gradient analysis between different training steps with the gradient analysis between different batches at initialization (Eq. 14). This is a good compromise given the restrictions of zero-shot NAS itself.
>
> Nevertheless, the empirical results show that our approximation by Eq. 14 can predict well the performance of neural networks and achieve better results than existing proxies. Hence, our proposed ZiCo analysis is a suitable approximation for the ideal gradient analysis where the convergence would be considered after certain training steps.
>
>
> **Q2 Secondly, it seems that the mathematical form of eq.14 can not be directly derived from Theorem 3.1 and 3.2, i.e., the form of coefficient of variation doesn't appear anywhere in these theorems. So, the motivation for using the form of eq.14 is kind of weak to me.**
>
> This is a very useful suggestion for us. We are currently not aware of any theoretical framework that would allow us to analyze the impact of the coefficient of variation directly on convergence or generalization (because most of the current tools assume a zero mean when looking at non-linear networks). We plan to do this in future work.
> As for the motivation, Eq. 14 is proposed by jointly considering the insights provided by Theorem 3.1 and 3.2. Essentially, Theorem 3.1 and Theorem 3.2 tell us that a good network should have high absolute mean values or low standard deviation (STD) values for the gradient.
>
> In practice, we found that using the mean or STD separately achieves worse correlation scores compared to Eq. 14. Specifically, we randomly select 2000 networks from NATSBench-SSS on CIFAR10, CIFAR100, and Img16-120 datasets and compute the following proxies: (i) Mean value of gradients only; (ii) Standard deviation (STD) value of gradients only; (iii) Combination of mean and std value, i.e., _ZiCo_. We then calculate the correlation coefficients between these proxies and the real test accuracy:
> |Proxy|C10(KT/SPR)|C100(KT/SPR)|Img16-120(KT/SPR)|
> |-|-|-|-|
> |Mean Only|0.25/0.37|0.39/0.55|0.61/0.81|
> |STD only|0.39/0.55|0.42/0.6|0.45/0.62|
> |**ZiCo(Mean+STD)**|**0.54/0.73**|**0.55/0.75**|**0.70/0.88**|
>
> In general, by considering the mean and STD values jointly, ZiCo (Eq. 14) achieves a much higher correlation than using each of them separately. This is why propose Eq. 14 to jointly consider both the mean and STD values of gradients. We provide more details in Appendix F.1 on Page 27.
>
>
> **Q3 Thirdly, Theorem 3.1 and 3.2 can only characterize the convergence of models whereas we need to characterize the generalization performance of architectures in NAS. So, is there any other justification that authors can provide to support why characterizing convergence is already enough?**
>
> We provide a new Theorem C.1 (Page 21) that links the gradient analysis with the test performance of neural networks. In principle, we prove that the networks with lower standard deviations of gradients have a better generalization capacity, thus having better test performance.  Hence, our proposed proxy can indicate both the convergence rate and generalization capacity of neural networks. We provide the proof and experimental results in Appendix C (Page 21).

---

> > ### Author Response · Authors · 2022-11-13
> > **Response to Reviewer 9avK (2/2)**
> >
> > **Q4 While this paper is motivated by the inconsistent performance of existing zero-shot proxies, the reason why the proposed method is able to tackle this problem has not been well explained either from the empirical perspective or the theoretical perspective.**
> >
> > In general, a good neural architecture should satisfy the following properties: good convergence/trainability and high generalization capacity. Existing proxies can indicate either convergence or generalization only. For example, SNIP and Synflow only indicate the trainability of neural networks. In contrast, as mentioned in the reply of Q3, ZiCo can indicate both the convergence and generalization capacity of neural networks. Therefore, ZiCo is more robust in finding a good architecture with both good convergence and generalization properties. We believe this is the reason why ZiCo consistently outperforms existing proxies. We'll revise the paper accordingly.
> >
> > **Q5 This paper misses the comparison with other training-free NAS works [GradSign, NASI] in their experiments.**
> >
> > In Table 4(Page 24), we provide new results of more comparisons between our proposed ZiCo and more baseline methods, namely: KNAS [1], NASWOT [2], GradSign[3], NTK on NATSBench (both NTK [4] and NASI [5] use NTK as the proxy but with different search algorithms).
> >
> > We compute the correlation scores on _NATSBench-TSS_ below:
> > |Proxy|C10(KT/SPR)|C100(KT/SPR)|Img16-120(KT/SPR)|
> > |-|-|-|-|
> > |KNAS[1]|0.14/0.2|0.24/0.35|0.3/0.42|
> > |NASWOT[2]|0.58/0.77|**0.62**/0.8|0.6/0.78|
> > |NTK(TE-NAS[4]/NASI[5])|0.33/0.44|0.33/0.43|0.46/0.63|
> > |GradSign[3]|0.58/0.77|0.59/0.79|0.59/0.78|
> > |**ZICO(ours)**|**0.61/0.8**|0.61/**0.81**|**0.6/0.79**|
> >
> > As shown above, ZiCo has a significantly higher correlation score than these proxies on NATSBench-TSS except NASWOT and GradSign.
> >
> > We also report correlation score _NATSBench-SSS_:
> > |Proxy|C10(KT/SPR)|C100(KT/SPR)|Img16-120(KT/SPR)|
> > |-|-|-|-|
> > |KNAS [1]|0.25/0.37|0.12/0.18|0.32/0.46|
> > |NASWOT [2]|0.45/0.63|0.43/0.59|0.42/0.59|
> > |NTK(TE-NAS[4]/NASI[5])|0.17/0.26|0.04/0.06|0.20/0.30|
> > |GradSign [3]|0.21/0.30|0.16/0.27|0.04/0.05|
> > |**ZICO(ours)**|**0.54/0.73**|**0.55/0.75**|**0.70/0.88**|
> >
> > As shown above, ZiCo has a significantly higher correlation score than these proxies on NATSBench-SSS.
> > Overall, our proposed ZiCo performs better than these proxies. We provide more details in Appendix E.1 (Page 25).
> >
> >
> > **Q6 While this paper is motivated by the inconsistent performance of existing zero-shot proxies, I recommend more search results on other diverse tasks (e.g., the ones mentioned in (White et al., 2022)) to support the improved consistency achieved by ZiCo.**
> >
> > We verify our proposed ZiCo on a new benchmark, TransNASBench-101-Micro[6]. We compare our proposed ZiCo with various proxies for the following four tasks: Scene Classification, Jigsaw, Autoencoding, and Surface Normal. We compute the correlation score between various proxies and the real test accuracy for these four tasks below:
> > |Proxy|Autoencoding(KT/SPR)|Scene Classification(KT/SPR)|Jigsaw(KT/SPR)|Surface Normal(KT/SPR)|
> > |-|-|-|-|-|
> > |Grad_norm|0.24/0.32|0.47/0.65|0.23/0.35|0.24/0.36|
> > |SNIP|0.20/0.27|0.52/0.71|0.27/0.41|0.32/0.49|
> > |Grasp|0.09/0.14|0.19/0.28|0.07/0.11|0.01/0.01|
> > |Fisher|**0.42/0.59**|0.49/0.67|0.19/0.30|0.10/0.14|
> > |Synflow|0.00/0.00|**0.53/0.72**|0.32/0.47|0.00/0.00|
> > |NASWOT|0.01/0.02|0.43/0.60|0.29/0.42|0.41/0.57|
> > |GradSign|0.01/0.02|0.32/0.46|**0.38/0.53**|0.29/0.40|
> > |Zen-score|0.09/0.14|0.52/0.72|0.35/0.50|**0.52/0.71**|
> > |Params|0.01/0.01|0.46/0.64|0.29/0.44|0.45/0.63|
> > |FLOPs|0.02/0.02|0.47/0.65|0.30/0.45|0.46/0.64|
> > |**ZiCo**|**0.24/0.35**|**0.51/0.71**|**0.36/0.52**|**0.50/0.68**|
> >
> > Clearly, our proposed ZiCo is consistently very close to the best scores except for Autoencoding. Though Fisher works better than ZiCo on Autoencoding, we note that ZiCo has a significantly higher score on the rest tasks. We note that existing proxies do not achieve a high correlation on all tasks consistently. In contrast, our proposed ZiCo achieves nearly the best results on all tasks except Autoencoding (we are the second best on Autoencoding). We provide more details in Appendix E.2 on Page 26.
> >
> > We hope that our response fully addresses Reviewer’s questions so Reviewer would consider raising the ratings for our work.
> >
> > [1] Xu, Jingjing, et al. "KNAS: green neural architecture search." ICML 2021.
> >
> > [2] Mellor, Joe, et al. "Neural architecture search without training." ICML 2021.
> >
> > [3] Zhang, Zhihao, et al. "GradSign: Model Performance Inference with Theoretical Insights." ICML 2021.
> >
> > [4] Chen, Wuyang, et al. "Neural Architecture Search on ImageNet in Four GPU Hours: A Theoretically Inspired Perspective." ICLR 2020.
> >
> > [5] Shu, Yao, et al. "NASI: Label-and Data-agnostic Neural Architecture Search at Initialization." ICLR 2021.
> >
> > [6] Duan, Yawen, et al. "Transnas-bench-101: Improving transferability and generalizability of cross-task neural architecture search." CVPR 2021.

---

> > > ### Author Response · Authors · 2022-11-16
> > > **Sincerely expecting further discussions with reviewer 9avK**
> > >
> > > Dear Reviewer 9avK:
> > >
> > > We want to thank you for the constructive comments in your review. As a follow-up to our responses, we would like to kindly remind you that the discussion period is ending soon. We hope to use this open response period to discuss the paper to solve the concerns and improve the quality of our paper. Have you gotten a chance to read our responses above, which attempt to address all of your concerns?
> > >
> > > We would be more than happy to provide more information or clarification and hope that they could lead to a positive and fair assessment of this paper.
> > >
> > > Best,
> > >
> > > Authors of paper 2989

---

> > > > ### Comment · Reviewer_9avK · 2022-11-22
> > > > **Thank you for the response**
> > > >
> > > > I thank the authors for the detailed response. I hope the authors can include all these discussions (especially for Q1, Q2 and Q3) in the revised paper. I would like to keep my score.

---

> > > > > ### Author Response · Authors · 2022-11-22
> > > > > **Thanks for the reply**
> > > > >
> > > > > Dear Reviewer 9avK,
> > > > >
> > > > >
> > > > > Thank you so much for participating in the discussion. We are happy to see that all your concerns were resolved. Based on your initial feedback, is there something else we can improve in our response that may help you increase the score for our paper? Please let us know.
> > > > >
> > > > >
> > > > > Thanks again,
> > > > >
> > > > >
> > > > > Authors of paper 2989

---

### Official Review · Reviewer_oG3W · 2022-10-25

**Confidence:** 4
**Correctness:** 3
**Technical Novelty And Significance:** 3
**Empirical Novelty And Significance:** 4
**Recommendation:** 6

**Clarity, Quality, Novelty And Reproducibility:**

Clearly written and contributes novel ideas. Overall good quality, should be easily reproducible with the provided code.

**Strength And Weaknesses:**

**Strengths:**

- a novel zero-cost proxy for NAS grounded in theory is always a nice thing to see
- overall good writing and organization of the paper
- good performance of the proposed method (although see below)

**Weaknesses:**

I haven't actually found anything that I would confidently consider a major weakness. I would have a series of suggestions how to improve writing in certain places, as well as possibly extending evaluation, but these are minor things.
More importantly, I have two questions to the authors which might turn out to be major weaknesses, but it is hard to conclude that just from reading a paper and therefore I would like to give the authors a chance to explain these things.

1. Theoretical contributions in Section 3 are concluded with a statement that "(...) network with high training convergence speed should
have high absolute mean values and low standard deviation values for the gradient (...)", which is used to motivate design of ZiCo.
I wouldn't say that this statement is incorrect (although, admittedly, the analysis from Section 3 it's not my primary area of expertise), however I have some doubts regarding applicability of this statement to NAS.
Specifically, it is a common pattern observed in practice that networks that converge faster usually converge to worse local minima - one of the most representatives examples could be two networks with significantly different number of parameters, usually a smaller network would converge much faster but to much worse results, whereas it takes longer to optimize networks with significantly more parameters but they tend to achieve better results.
How is the analysis provided helpful in those situations? Does it provide us with means to correctly identify such cases?
Additional bonus question: doesn't the assumption about $g_i~\mathcal{N}(0,\sigma)$ in Theorem 3.2 undermine applicability of Theorem 3.1? If I am not mistaken, under this assumption $\mu_j$ in Eq. 5 is going to be 0, and Remark 3.1 is not really helpful.


2. How much ImageNet results depend on distillation from EfficientNet-B3?
Personally, I consider it cheating when a NAS methods shows that it can discover "better" architectures but compares them using a different training scheme.
It is a well-known fact that even old architectures can benefit greatly from utilizing state-of-the-art training scheme (e.g., see "ResNet strikes back" by R. Wightman et al.), and it seems that the paper currently contains comparisons between ZiCo and other NAS methods where ZiCo might have unfair advantage of an improved training scheme.
For example, the paper makes direct comments about results such as: "Moreover, if the FLOPs is 600M, ZiCo achieves 2.6% higher Top-1 Accuracy than the latest one-shot NAS method (MAGIC-AT) with a 3× reduction in terms of search time" - not only is MAGIC-AT not the best Zero-Cost method for 600M FLOPs in Table 3, the authors of MAGIC-AT specifically write in their paper: "For stand-alone model training, to be consistent with the previous works, we follow the same strategy as ProxylessNAS5 and do not employ tricks like cutout or mixup", while ZiCo not only includes all these tricks, it also employs distillation from much more accuracy EfficientNet-B3!
On the other hand, ZenNAS, which roughly follows the same training procedure, achieves much closer results to the proposed method, up to the point that it might very well lie within randomness of each method.
To adequately address this issue, I would suggest the authors to at least limit their comparison to methods that use the same search space and training scheme, but ideally also include NAS results in a different setting, as presenting end-to-end results only on a single search space is, in general, considered rather limited. How about performance of the proposed method on a significantly different search space like DARTS?
Also, how about comparison with other zero-cost methods?
In general, my current impression is that the choice of baselines and retraining scheme was a bit cherry-picked to present the proposed method in a favourable way.
I am happy to be convinced it is not the case, though.

**Minor suggestions and questions:**

 - within your computational capabilities, please try to run experiments multiple times and report avg. + std. (or whatever other
confidence metric)
 - is there a point in highlighting "key questions" and "major contributions" in Introduction? Is is there to make sure the reader understands that the contributions are major or that the following bullet points are in fact contributions?
 - "We demonstrate that, compared to all existing proxies (...)" - results in the paper hardly demonstrate comparison with "_all_ existing proxies"... I would expect more linguistic rigour from a math-heavy paper like this; examples of published proxies that are not included in the comparison: NASWOT (Mellor et al., ICLM'21), TE-NAS (Chen et al., ICLR'21), KNAS (Xu et al., ICML'21), NASI (Shu et al., ICLR'22), GradSign (Zhang and Jia, ICLR'22).
 - NATSBench-TSS is not NASBench201, as suggested by the header in Table 1
 - how would you explain the fact that correlation of ZiCo degrades with more batches? Shouldn't more batches result in better estimates of mean and variance, resulting in better performance of the proxy?
 - I am not exactly sure if highlighting that the proposed method works consistently better than #Params on benchmarks where #Params happens to be highly correlated with performance is really important for the paper... To be clear, it is worth mentioning that the proposed method works the best, etc., but specifically highlighting superiority w.r.t. the #Params baseline is a bit too narrow in my opinion - what about cases when #Params is not a good proxy? (the comment is specifically related to contribution 2)


**Summary Of The Paper:**

The paper proposes a new zero-cost proxy (called ZiCo) for neural architecture search motivated by theoretical insights about the relationship of statistics of gradient across different input samples (absolute mean and standard deviation) and a network's converge speed.
The proposed proxy is evaluated by investigating ranking correlation on NATSBench-TSS, NATSBench-SSS and NASBench101, as well as by end-to-end NAS performance (guiding an evolutionary algorithm) using MobileNetv2 search space on ImageNet.
Additionally, experiments designed to validate theoretical claims and ablations, including investigation of ZiCo's correlation to accuracy on NATSBench-TSS with varying number of batches and batch size, are also presented.

**Summary Of The Review:**

I like the paper and the presented attempt at designing a theoretically-grounded zero-cost proxy for NAS.
Overall, I am leaning towards accepting the paper even with a high score. However, since I am not sure about some important aspects of the paper, I am leaving a lowered score for now and hope that the authors will be able to adequately address my concerns.

---

> ### Author Response · Authors · 2022-11-13
> **Response to Reviewer oG3W (1/3)**
>
> **Q1 It is a common pattern observed in practice that networks that converge faster usually converge to worse local minima: usually a smaller network converges faster but to worse results, whereas it takes longer to optimize networks with more parameters but to better results. How is the analysis provided helpful in those situations?**
>
> We provide a new Theorem C.1 (Page 21) that links the gradient analysis with the test performance of neural networks. In principle, we prove that the networks with a lower standard deviation of gradients also have a better generalization capacity, thus a better test performance.  Hence, the networks with higher ZiCo scores have not only a faster convergence rate, but also better test accuracy. This way, ZiCo-based zero-shot NAS can avoid finding the network with poor accuracy. We provide the proof and experimental results in Appendix C (Page 21).
>
> **Q2 Doesn't the assumption about $g_i$ $N(0,σ)$ in Theorem 3.2 undermine applicability of Theorem 3.1? If I am not mistaken, under this assumption $μ_j$ in Eq. 5 is going to be 0, and Remark 3.1 is not really helpful.**
>
> We remark that Theorem 3.1 has been proved for a linear network, while Theorem 3.2 has been proved for a nonlinear network, i.e., an MLP with ReLU activation functions. Therefore, these two theorems should be considered separately since they have different prerequisites.
>
> Moreover, we note that the theoretical analysis of non-linear networks is very difficult.  To conduct the analysis of ReLU MLP, our proof of Theorem 3.2 uses knowledge from kernel-based theoretical tools; we haven’t found any theoretical framework that could help us analyze the mean values of gradients. Hence, in Theorem 3.2, we don’t consider the mean values of gradients for the non-linear case. We will try to extend our results for the non-linear case in our future work.
>
> Nevertheless, we find some empirical evidence that shows that the mean values are also related to the performance of neural networks with ReLU. Specifically, we explore the following proxies: (i) Mean value of gradients only; (ii) Standard deviation (STD) value of gradients only; (iii) Combination of mean and STD value, i.e., _ZiCo_. We then calculate the correlation scores between these proxies and the test accuracy on NATSBench-SSS where the networks are all using ReLU as activation functions:
> |Proxy| C10 (KT/SPR)|C100 (KT/SPR)|Img16-120 (KT/SPR)|
> |-|-|-|-|
> |Mean Only|0.25/0.37|0.39/0.55|0.61/0.81|
> |STD only|0.39/0.55|0.42/0.6|0.45/0.62|
> |**ZiCo (Mean+STD)**|**0.54/0.73**|**0.55/0.75**|**0.70/0.88**|
>
> As shown above, both mean only and STD only are correlated with the performance of networks. Moreover, by combining mean and STD together, ZiCo achieves a much higher correlation than using each of them individually. We provide more details in Appendix F.1 on Page 27.
>
> **Q3 How much ImageNet results depend on distillation from EfficientNet-B3? I would suggest the authors to at least limit their comparison to methods that use the same search space and training scheme.**
>
>
> We agree that it’s very important to make a fair comparison, and hence we put our best efforts in the initial submission. Let us explain: First, our search space is the same as the space used by the most popular NAS methods (MobileNet-v2+SE) [1,2,3,4,5,6,7]. Second, as for the training setup, the difficulty we have is that different NAS papers usually use different training setups. Moreover, it is not practical to rerun all the NAS papers under the same setup due to the limits of the computation capacity. To address this issue, we consider the following two things:
> - We compare ZiCo with existing proxies on multiple NAS benchmarks since these benchmarks can provide a fair setup for the comparison (pls. check the results in Tables 4/5/6 on Page 24/25/26).
> - We select the training method of ZenNAS; this is because, to our best knowledge, TE-NAS and ZenNAS are the only two zero-shot NAS approaches that report results on ImageNet. ZenNAS is also the previous SOTA zero-shot NAS method.
>
> To further address the reviewer’s concern, we train the network obtained via ZiCo under the same setup as MAGIC-AT [7]. Specifically, we train the network with 150 epochs _without_ any advanced data augmentation techniques (e.g., mixup, RandAugment, or cutout) and _without_ knowledge distillation. We also train ZenNAS and EfficientNets under the same setup. The results on shown below:
> |Method|-|-|Costs(GPU days)|
> |-|-|-|-|
> |Zen-score| 75.6(410M)|76.1(611M)|0.5|
> |EfficientNet|76.0(390M)|77.4(700M)|3800|
> |ZiCo|76.5$\pm$0.2(448M)|77.1$\pm$0.3(603M)|0.4|
>
> Overall, we can see that without knowledge distillation, ZiCo still outperforms the previous zero-shot NAS approaches. Moreover,  compared to the one-shot and multi-shot NAS methods, ZiCo achieves a higher or comparable accuracy with much less search time. We provide more comparisons in Table 9 on Page 28.

---

> > ### Author Response · Authors · 2022-11-13
> > **Response to Reviewer oG3W (2/3)**
> >
> > **Q4 I would suggest the authors ideally also include NAS results in a different setting, as presenting end-to-end results only on a single search space is, in general, considered rather limited. How about performance of the proposed method on a different search space like DARTS?**
> >
> > We note that ZiCo has been verified on multiple search spaces, e.g., NATSBench-TSS, NATSBench-SSS, NASBench101, and MobileNet-v2+SE search space with up to 12 different proxies. These search spaces are quite different from each other. For example, NATSBench-TSS contains architectures with different cell structures, which is very similar to DARTS search space; NATSBench-SSS consists of networks that have the same operation and differ only in the width of operations.
> >
> > Moreover, we conducted new experiments on the following new NAS benchmarks, _TransNAS-Bench-101-Mirco_, which cover an entirely new search space. It also consists of pretty diverse applications, including image classification, pixel-level prediction, and image reconstruction (check Appendix E.2 on Page 26). As shown in Table 4 (Page 24), Table 5(Page 25), and Figure 2(Page 7), in most of the search spaces and most of the tasks, ZiCo achieves the best performance compared to the existing zero-shot proxies.
> >
> > To further address the reviewer’s concern, we use ZiCo to conduct the search within the DARTS search space on CIFAR10 and datasets. The results are shown below:
> >
> > |Approach|TestError(%)|Method|Cost(GPUdays)|
> > |-|-|-|-|
> > |ENAS|2.89|MS|0.5|
> > |NASNet-A|2.65|MS|2000|
> > |DARTS|2.76$\pm0.09$|OS|1|
> > |SNAS|2.85$\pm0.02$|OS|1.5|
> > |BayesNAS|2.81$\pm0.04$|OS|0.2|
> > |ProxylessNAS|2.08|OS|4|
> > |PC-DARTS|2.57$\pm0.07$|OS|0.1|
> > |SDARTS-ADV|2.61$\pm0.02$|OS|1.3|
> > |Zen-score|2.55$\pm0.04$|ZS|0.01|
> > |TE-NAS|2.63$\pm0.064$|ZS|0.05|
> > |ZiCo(ours)|2.45$\pm0.11$|ZS|0.03|
> >
> > As shown above, ZiCo outperforms previous zero-shot NAS approaches, e.g, Zen-score and TE-NAS. Moreover, compared to the regular one-shot or multi-shot NAS methods, ZiCo achieves comparable or higher test accuracy with at least 10$\times$ less search time.
> >
> >
> > **Q5 Within your computational capabilities, please try to run experiments multiple times and report avg. + std. (or whatever other confidence metric)**
> >
> > We appreciate this suggestion and have updated Table 3 accordingly. Specifically, we have run the experiments 3 times for our method and reported the mean +/- std.
> >
> >
> > **Q6 I would expect more linguistic rigour from a math-heavy paper like this; examples of published proxies that are not included in the comparison: NASWOT, TE-NAS, KNAS, NASI, GradSign.**
> >
> > In Table 4 (Page 24), we provide new results of more comparisons between our proposed ZiCo with these methods: NASWOT [8], GradSign [9], KNAS [12], and NTK on NATSBench (both NTK [10]  and NASI [11] use NTK as the proxy but with different search algorithms).
> >
> > We compute the correlation score between these proxies vs. test accuracy on _NATSBench-TSS_:
> > |Proxy|C10(KT/SPR)|C100(KT/SPR)|Img16-120(KT/SPR)|
> > |-|-|-|-|
> > |KNAS [12]|0.14/0.2|0.24/0.35|0.3/0.42|
> > |NASWOT [8]|0.58/0.77|**0.62**/0.8|0.6/0.78|
> > |NTK(TE-NAS[10]/NASI[11])|0.33/0.44|0.33/0.43|0.46/0.63|
> > |GradSign [9]|0.58/0.77|0.59/0.79|0.59/0.78|
> > |**ZICO(ours)**|**0.61/0.8**|**0.61/0.81**|**0.6/0.79**|
> >
> > As shown above, ZiCo has a significantly higher correlation score compared to these proxies on NATSBench-TSS except NASWOT and GradSign.
> >
> > We also report correlation score on _NATSBench-SSS_:
> > |Proxy|C10(KT/SPR)|C100(KT/SPR)|Img16-120(KT/SPR)|
> > |-|-|-|-|
> > |KNAS [12]|0.25/0.37|0.12/0.18|0.32/0.46|
> > |NASWOT [8]|0.45/0.63|0.43/0.59|0.42/0.59|
> > |NTK(TE-NAS[10]/NASI[11])|0.17/0.26|0.04/0.06|0.20/0.30|
> > |GradSign [9]|0.21/0.30|0.16/0.27|0.04/0.05|
> > |**ZICO(ours)**|**0.54/0.73**|**0.55/0.75**|**0.70/0.88**|
> >
> > As shown above, ZiCo has a consistently higher correlation score than these proxies on NATSBench-SSS.
> > Overall, our proposed ZiCo performs better than these proxies. We provide more details in Appendix E.1 on Page 25.

---

> > > ### Author Response · Authors · 2022-11-13
> > > **Response to Reviewer oG3W (3/3)**
> > >
> > > **Q7 is there a point in highlighting "key questions" and "major contributions" in Introduction? Is it there to make sure the reader understands that the contributions are major or that the following bullet points are in fact contributions?**
> > >
> > > With the "key questions" and "major contributions", our goal was to precisely highlight what problem we are addressing and to summarize our main contributions. Moreover, as clearly evident from all of our results, our contributions are indeed major to the field of zero-shot NAS. This is why we presented our contributions this way. We hope that this clarifies the issue.
> > >
> > >
> > > **Q8 NATSBench-TSS is not NASBench201, as suggested by the header in Table 1**
> > >
> > > At the official website of the NATSBench, the authors say that they are the same: ‘The Topology Search Space in NATS-Bench is the same as NAS-Bench-201’. (pls. check the paragraph on top of _Preparation and Download_ in [NATS-Github](https://github.com/D-X-Y/NATS-Bench#preparation-and-download))
> > >
> > > **Q9 how would you explain the fact that correlation of ZiCo degrades with more batches? Shouldn't more batches result in better estimates of mean and variance, resulting in better performance of the proxy?**
> > >
> > > We are not exactly sure of the reason behind this degradation. One possibility is the assumption of our theoretical analysis: our Lemma 1 requires the width of the network $m$ satisfies $m=\Omega(\frac{M^6}{\lambda_0^4\delta^3})$, i.e., $M=\Omega(\sqrt[6]{m\lambda_0^4\delta^3})$, where $M$ is the number of training samples, $\lambda_0$ and $\delta$ are some positive constants. Hence, there may exist a suitable range of the values of $M$ where ZiCo can indicate the convergence and accuracy of networks. When the number of batches increases, the value of $M$ (the number of training samples) may become too large and thus are beyond the suitable range. To empirically verify this, we increase the batch size gradually with two batches and then compute the correlation scores for 2000 networks from NATSBench-TSS on the CIFAR100 dataset:
> > >
> > > |Batch size|16|64|256|1024|4096|
> > > |-|-|-|-|-|-|
> > > |Kendall’s $\tau$|0.586|0.596|0.598|0.589|0.564|
> > >
> > > As shown above, there’s a slight degradation when the batch size is too small or too big (e.g., 16 or 4096), which coincides with our above conjecture. We also note that, when applying ZiCo in practice, the degradation in correlation score is negligible(at most 0.03) so it doesn’t matter much.
> > >
> > > **Q10 I am not exactly sure if highlighting that the proposed method works consistently better than #Params on benchmarks where #Params happens to be highly correlated with performance is really important for the paper... To be clear, it is worth mentioning that the proposed method works the best, etc., but specifically highlighting superiority w.r.t. the #Params baseline is a bit too narrow in my opinion - what about cases when #Params is not a good proxy? (the comment is specifically related to contribution 2)**
> > >
> > > We appreciate the reviewer’s suggestion. We have modified Contribution 2 to make the statement broader. We also note that #Params is not the only baseline method compared with our Proposed ZiCo. As shown in Table 4 (Page 24), we have compared ZiCo against 12 baseline proxies, in total. Hence, we rephrased Contribution 2 accordingly.
> > >
> > > We hope that our response fully addresses Reviewer’s questions so Reviewer would consider raising the ratings for our work.
> > >
> > > [1] Pham, Hieu, et al. "Efficient neural architecture search via parameters sharing." ICML 2018.
> > >
> > > [2] Lin, Ming, et al. "Zen-nas: A zero-shot nas for high-performance image recognition." ICCV 2021.
> > >
> > > [3] Cai, Han, et al. "Once-for-all: Train one network and specialize it for efficient deployment." ICLR 2020.
> > >
> > > [4] Yu, Jiahui, et al. "Bignas: Scaling up neural architecture search with big single-stage models." ECCV 2020.
> > >
> > > [5] Cai, Han, et al. "Proxylessnas: Direct neural architecture search on target task and hardware." ICLR 2019.
> > >
> > > [6] Tan, Mingxing, et al. "Efficientnet: Rethinking model scaling for convolutional neural networks." ICML 2019.
> > >
> > > [7] Xu, Jin, et al. "Analyzing and mitigating interference in neural architecture search." ICML 2022.
> > >
> > > [8 ] Mellor, Joe, et al. "Neural architecture search without training." ICML 2021.
> > >
> > > [9] Zhang, Zhihao, et al. "GradSign: Model Performance Inference with Theoretical Insights." ICML 2021.
> > >
> > > [10] Chen, Wuyang, et al. "Neural Architecture Search on ImageNet in Four GPU Hours: A Theoretically Inspired Perspective." ICLR 2020.
> > >
> > > [11] Shu, Yao, et al. "NASI: Label-and Data-agnostic Neural Architecture Search at Initialization." ICLR 2021.
> > >
> > > [12] Xu, Jingjing, et al. "KNAS: green neural architecture search." ICML 2021.

---

> > > > ### Author Response · Authors · 2022-11-16
> > > > **Sincerely expecting further discussions with reviewer oG3W**
> > > >
> > > > Dear Reviewer oG3W:
> > > >
> > > > We want to thank you for the constructive comments in your review. As a follow-up to our responses, we would like to kindly remind you that the discussion period is ending soon. We hope to use this open response period to discuss the paper to solve the concerns and improve the quality of our paper. Have you gotten a chance to read our responses above, which attempt to address all of your concerns?
> > > >
> > > > We would be more than happy to provide more information or clarification and hope that they could lead to a positive and fair assessment of this paper.
> > > >
> > > > Best,
> > > >
> > > > Authors of paper 2989

---

> > > > ### Comment · Reviewer_oG3W · 2022-12-01
> > > > **Addressing rebuttal**
> > > >
> > > > I would like to thank the authors for their response - overall I am happy to recommend acceptance, although there is still room for improvement in my opinion (please see the details below).
> > > >
> > > > Regarding generalization, I think the extra analysis included by the authors is interesting and potentially useful for NAS community, but at the same time misses the point a little bit.
> > > > Basically, my source of scepticism in that regard is related to practical applicability of the theoretical insights - although we see that empirical results of ZiCo are good (now with extra experiments without KD and on the DARTS-C10 this part in particular is much more convincing), the paper still does not really address the issue of the discrepancy between convergence speed and final performance which we can often see among networks used in practice (as opposed to the two-layer MLPs trained on MNIST in C.3 - and even when such networks are used, the correlation showed in Figure 4 is not exactly convincing to me, the clearly visible trend seems to only exist for large standard deviations, but especially if we focus on the very best models, they do not seem to consistently have small $\sigma$).
> > > > What I would imagine to be an adequate attempt at addressing this issue, would be to take a couple of practical networks that we know converge at different rates to different results and see if ZiCo is capable of correctly scoring them, this could be some older or newer SOTA models, or models from a NAS benchmark that includes training curves.
> > > >
> > > > If ZiCo does score them correctly, then that's great but the authors would probably need to reconsider their implicit statement about the relationship between convergence speed and final performance (that would basically mean that ZiCo does something else than focus on converge speed, which goes against the motivation presented in the paper); if it does not work then the relationship is somewhat consistent but there is an important limitation in what ZiCo can do - I wouldn't consider this to undermine the paper by any means, but on the other hand not mentioning this at all would look very deceptive.
> > > > In either case, the problem reduces to the fact that some important limitations of the method and/or reasoning behind ZiCo's design are not addressed. As a matter of fact, the paper does not seem to discuss limitations of any of its parts at all, which in some sense proves my point.
> > > >
> > > > The above is really the main point (please note that "minor" points from my original review are of secondary significance). In the rest of the post I will simply comment on the rest of the response in case it provides useful information for the authors to further polish their paper.
> > > >
> > > > Re. Q2: I think the empirical results are more than enough here - perhaps would be good to mention in the paper that despite T3.1 and T3.2 being seemingly mutually exclusive due to their assumptions, we can empirically see that considering both mean and std at the same time is beneficial?
> > > >
> > > > Re. Q3: The results are enough to show that ZiCo indeed finds good archs, but please tweak your writing to avoid directly comparing to methods that use different training scheme (i.e., the sentence from my original review) - I understand that according to Table 9 you are still better than MAGIC-AT, but 1) it's not by 2.6%, and 2) this comparison is presented deep in the Appendix, while in the main text the Table contains results that should not be compared directly. Again, this does not undermine the results but please iron out details like this.
> > > >
> > > > Re. Q4: Regarding your "note" about multiple search spaces, please note that my comment was about end-to-end NAS results, not correlations. I'm happy with the extra experiments and I can see more NAS results in the revised paper, but originally it was basically just MobileNet and NB2.
> > > > Also, saying that NB2 "is very similar to DARTS search space" is a big stretch...
> > > >
> > > > Re. Q6: I appreciate comparisons with more proxies, although please note that I was merely making a comment about your writing - I understand this is basically nitpicking from my side, but please avoid saying things like "all previously proposed proxies" (still there at the end of section 4.1) - why not just say "all <number of proxies> of the baseline proxies considered in this paper"?
> > > >
> > > > Re. Q7: again, this was merely a comment about highlighting these two phrases in bold. In my opinion, drawing a reader's attention (that is what bold font does, after all) specifically to these phrases has little benefit. If I wanted to go through your paper really quickly, I would rather appreciate bolding the parts that actually contain information about the content of your paper, e.g. "a new theoretically-grounded proxy", etc.
> > > > Still, this is again just nitpicking from my side, so it completely up to your preference.
> > > >
> > > > Re. Q8: Apologies for the mistake, I was thinking about NATSBench as a whole while making this comment, but you're right the TSS parts is NB201.

---

> > > > > ### Comment · Reviewer_oG3W · 2022-12-01
> > > > > **Missing comment**
> > > > >
> > > > > Seems like I forgot to mention one more thing: in some of the new results (e.g., Table 5, 8) NASWOT is attributed to Lopes et. al (2021). I'm not sure if this is simply an error in citation or if the authors really mean the proxy proposed in this work (it does propose a metric derived from the old naswot score, but it's a different thing from NASWOT score used in the ICML version which is cited in other places, e.g. Table 4). In either way it would need to be fixed.

---

> > > > > > ### Author Response · Authors · 2022-12-06
> > > > > > **Thank you for constructive feedback (1/2)**
> > > > > >
> > > > > > Dear Reviewer oG3W,
> > > > > >
> > > > > > We appreciate your time and feedback on our responses and thank you so much for raising our score.
> > > > > >
> > > > > > Regarding your newest feedback, please see our responses below:
> > > > > >
> > > > > > **Convergence and Test Performance** We thank you for the great suggestions. Indeed, the analysis of practical networks is a great idea to better illustrate the potential of ZiCo. To this end, we analyze the correlation between ZiCo scores and the training loss using the NASBench-201 benchmark after 12 and 200 training epochs (since NASBench-201 only provides the training information for these two epochs).  Our results are given below:
> > > > > >
> > > > > >
> > > > > > _ZiCo scores vs. Training loss (Epoch 12):_
> > > > > > |Dataset|CIFAR10|CIFAR100|Img16-120|
> > > > > > |-|-|-|-|
> > > > > > |Kendall's $\tau$|0.44|0.38|0.40|
> > > > > > |Spearman's $\rho$|0.58|0.55|0.57|
> > > > > >
> > > > > >
> > > > > >
> > > > > > _ZiCo scores vs. Test Accuracy (Epoch 12):_
> > > > > > |Dataset|CIFAR10|CIFAR100|Img16-120|
> > > > > > |-|-|-|-|
> > > > > > |Kendall's $\tau$|0.41|0.37|0.37|
> > > > > > |Spearman's $\rho$|0.57|0.53|0.52|
> > > > > >
> > > > > >
> > > > > > _Training loss (Epoch 12) vs. Test Accuracy (Epoch 12):_
> > > > > > |Dataset|CIFAR10|CIFAR100|Img16-120|
> > > > > > |-|-|-|-|
> > > > > > |Kendall's $\tau$|0.96|0.95|0.92|
> > > > > > |Spearman's $\rho$|0.99|0.99|0.99|
> > > > > >
> > > > > >
> > > > > > _ZiCo scores vs. Training loss (Epoch 200):_
> > > > > > |Dataset|CIFAR10|CIFAR100|Img16-120|
> > > > > > |-|-|-|-|
> > > > > > |Kendall's $\tau$|0.61|0.66|0.61|
> > > > > > |Spearman's $\rho$|0.80|0.86|0.80|
> > > > > >
> > > > > >
> > > > > >
> > > > > >
> > > > > > _ZiCo scores vs. Test Accuracy (Epoch 200):_
> > > > > > |Dataset|CIFAR10|CIFAR100|Img16-120|
> > > > > > |-|-|-|-|
> > > > > > |Kendall's $\tau$|0.61|0.61|0.60|
> > > > > > |Spearman's $\rho$|0.80|0.81|0.79|
> > > > > >
> > > > > >
> > > > > > _Training loss (Epoch 200) vs. Test Accuracy (Epoch 200):_
> > > > > > |Dataset|CIFAR10|CIFAR100|Img16-120|
> > > > > > |-|-|-|-|
> > > > > > |Kendall's $\tau$|0.84|0.80|0.90|
> > > > > > |Spearman's $\rho$|0.96|0.94|0.98|
> > > > > >
> > > > > >
> > > > > > _Training loss (Epoch 12) vs. Test Accuracy (Epoch 200):_
> > > > > > |Dataset|CIFAR10|CIFAR100|Img16-120|
> > > > > > |-|-|-|-|
> > > > > > |Kendall's $\tau$|0.70|0.64|0.67|
> > > > > > |Spearman's $\rho$|0.87|0.82|0.84|
> > > > > >
> > > > > >
> > > > > > Based on the above results, we make the following observations on NASBench-201:
> > > > > > - ZiCo scores are good at predicting both training convergence and testing performance. We will revise the paper to reflect the new convergence and generalization results, even though predicting training convergence is slightly better than the final test performance.
> > > > > > - ZiCo scores have a higher correlation with training loss and test accuracy as the number of training epochs increases (12 epochs vs. 200 epochs). We think this is because at the early stages of training, the learning rate is relatively high, thus there’s a higher noise for both convergence and test performance.
> > > > > > - There’s a high correlation score (larger than 0.64) between training loss and test accuracy. These results show that, in general, convergence rate correlates well with the test performance in the NASBench-201 search space.
> > > > > >
> > > > > > Besides NASBench-201, we also conducted experiments with the ResNet family on the Cifar10 dataset. We train these networks with 200 epochs and set the initial learning rate as 0.1 and reduce it by a factor of 10 at epoch [100,150]. The results are given below (as averages over three runs):
> > > > > >
> > > > > > |Network|ZiCo|Training loss@Epoch1|Test accuracy@Epoch1|Test accuracy@Epoch200|
> > > > > > |-|-|-|-|-|
> > > > > > |ResNet20|221|1.66|39.75|91.73|
> > > > > > |ResNet32|361|2.18|22.57|92.63|
> > > > > > |ResNet44|506|2.31|19.4|93.1|
> > > > > > |ResNet56|632|2.71|13.0|93.39|
> > > > > > |ResNet110|1263|2.89|10.05|93.68|
> > > > > >
> > > > > > As shown above, there’s no clear relationship between the final test accuracy and initial training convergence for the ResNet family. To analyze the training convergence vs. generalization in more detail, we also plot the training/testing accuracy curves for ResNet20 and ResNet110 in the [anonymous link](https://drive.google.com/drive/folders/1Wy5hlQ2c1wnusPdiV669krAuskWU9ntd?usp=share_link). Clearly, the results show that the reviewer’s comment is correct for the ResNet family; that is, the smaller ResNet (ResNet20) initially converges faster, but achieves lower final test accuracy than the larger ResNet (ResNet110). Most importantly, for this case, ZiCo correctly scores the two networks in terms of final test accuracy.
> > > > > >
> > > > > > By jointly considering the results on NASBench-201 and ResNet, we believe the relationship between the convergence and generalization depends on the search space. Currently, ZiCo doesn’t take the features of the search space into account, which is a limitation we plan to explore in future work. Moreover, we plan to improve in another direction, namely to relax the assumption of zero means in the theoretical analysis of ReLU networks. In the final version of this paper, we will add a new section listing these current limitations, as suggested.

---

> > > > > > ### Author Response · Authors · 2022-12-06
> > > > > > **Thank you for constructive feedback (2/2)**
> > > > > >
> > > > > > **Citation of NASWOT**: We appreciate the reviewer’s catch. For Table  5 and Table 8, the correct cited paper should be [1]. We will fix this citation in the final version of the paper.
> > > > > >
> > > > > >
> > > > > > **Q2** We will add the following sentence to Section 3.3  in the final version of our paper:
> > > > > >
> > > > > > _Despite Theorem 3.1 being mutually exclusive with Theorem 3.2 and Theorem 3.3 (due to their assumptions), we empirically show (see Table 7) that by jointly considering both mean and std deviations, we achieve better correlation scores compared to using them individually._
> > > > > >
> > > > > >
> > > > > >
> > > > > > **Q3** We will merge Table 3 and Table 9 and update the following sentence to Section 5.4 in the final version of our paper:
> > > > > >
> > > > > > _Moreover, if the FLOPs is 600M, ZiCo achieves 0.3% higher Top-1 Accuracy compared to  the latest one-shot NAS method (MAGICAT) with a 3× reduction in terms of search time_
> > > > > >
> > > > > >
> > > > > > **Q4**: We appreciate the reviewer's question which helps us further improve our paper.
> > > > > >
> > > > > >
> > > > > > **Q6** In the final version, we will update the following sentence in Section 4.1:
> > > > > >
> > > > > > _In practice, two batches are enough to make ZiCo achieve the SOTA performance among 12 existing representative accuracy proxies (see Sec. 5.5)._
> > > > > >
> > > > > >
> > > > > > **Q7**: We thank you for these suggestions; we will update the paper accordingly in the final version.
> > > > > >
> > > > > >
> > > > > >
> > > > > >
> > > > > > Thank again for your feedback and best wishes,
> > > > > >
> > > > > > Authors of paper 2989
> > > > > >
> > > > > > [1] Mellor, Joe, et al. "Neural architecture search without training." International Conference on Machine Learning, 2021.

---

### Official Review · Reviewer_V9fp · 2022-10-28

**Confidence:** 4
**Correctness:** 3
**Technical Novelty And Significance:** 3
**Empirical Novelty And Significance:** Not applicable
**Recommendation:** 8

**Clarity, Quality, Novelty And Reproducibility:**


Question about the theorem 3.1 and theorem 3.3:

1.Theorem 3.1 and Theorem 3.3 tell us that the network with low training loss should have high absolute mean values and low standard deviation values for the gradient. So why not just use the training loss item as a proxy?

2.The validation experiment of the theorem 3.1 and theorem 3.3 focus on the training loss. I think it is more convincing to explore the correlation between them and the real accuracy on the validation set.

3.It seems that the caption of Figure 1 draws the wrong opposite conclusion.

Question about the evaluation on NAS benchmark and Imagenet:

1.I think the recent zero-cost/zero-shot literatures should be considered and compared. E.g., the zero-cost PT showing a best architecture of 2.43 top1 error on CIFAR-10, while only using 0.018 hours search cost.

**2.Why only show the result of Zen-score partly, ignoring its better results of 83.6% under the 1000M FLOPS constraint?**

3.How about the results without distillation of Efficient-B3?

4.Since the proxy ZiCo is defined from the absolute mean values and standard deviation values, it will be interesting to explore their individual contribution.



[a] Zero-Cost Proxies Meet Differentiable Architecture Search, Xiang et al.


**Strength And Weaknesses:**


Strength:

The authors theoretically reveal how the mean value and standard deviation of gradients impact the convergence of neural networks. And based on their theoretical analysis, they propose a new train-freet proxy, ZiCo.  Experiment results have shown that ZiCo works better than previous zero-shot NAS proxies on multiple popular NAS-Benchmarks (NASBench101, NATSBench-SSS/TSS) for multiple datasets (CIFAR10/100, ImageNet16-120).

Weaknesses:

Several concerns in  **Clarity, Quality, Novelty, Reproducibility**.

**Summary Of The Paper:**

The paper first theoretically reveal how some specific gradient properties impact the convergence rate of neural networks. Based on their theoretical analysis, they propose a new zero-shot proxy, *ZiCo*, the first train-free proxy that works consistently better than #Params.


**Summary Of The Review:**


The authors theoretically reveal how the mean value and standard deviation of gradients impact the training loss of neural networks. And based on their theoretical analysis, they propose a new train-free proxy, ZiCo, which achieves better correlation scores with the real test accuracy. But there are still several concerns need to be addressed.

# Update after reading response

I think all my previous concerns are well addressed. Thanks the author for their effort. I raise my score as promised.

---

> ### Author Response · Authors · 2022-11-13
> **Response to Reviewer V9fp (1/2)**
>
> **Q1 Theorem 3.1 and Theorem 3.3 tell us that the network with low training loss should have high absolute mean values and low standard deviation values for the gradient. So why not just use the training loss item as a proxy?**
>
> There are two reasons for not using the training loss as a proxy directly: On the one hand, for an initialized network, without training, the training loss value will not change and remain a random number. Hence, it cannot be used to do zero-shot NAS. On the other hand, since our method targets zero-shot NAS, network training is strictly forbidden. Thus we cannot use training loss as a proxy in zero-shot NAS. We also note that the gradients used in our proxy are all computed based on the initialized network, i.e., without updating the parameters.
>
>
> **Q2 The validation experiment of the theorem 3.1 and theorem 3.3 focus on the training loss. I think it is more convincing to explore the correlation between them and the real accuracy on the validation set.**
>
> We provide a new Theorem C.1 (Page 21) that links the gradient analysis with the test performance of neural networks. To summarize, we prove that the networks with lower standard deviations of gradients also have a better generalization capacity, and thus better test performance.  Hence, our proposed proxy can indicate both the convergence rate and generalization capacity of neural networks. We provide the details of the proof and experimental results in Appendix C (Page 21).
>
>
> **Q3 It seems that the caption of Figure 1 draws the wrong opposite conclusion.**
>
> We appreciate the reviewer’s great catch; we have corrected this caption accordingly.

---

> > ### Author Response · Authors · 2022-11-13
> > **Response to Reviewer V9fp (2/2)**
> >
> >
> > **Q4 I think the recent zero-cost/zero-shot literatures should be considered and compared. E.g., the zero-cost PT**
> >
> > We remark that zero-cost PT [1] is a search algorithm rather than a zero-shot proxy; hence our proposed proxy is orthogonal to zero-cost PT. In principle, all zero-shot proxies can be combined with zero-cost PT. To illustrate this, we implement the zero-cost PT algorithm by ourselves (since the official code is not released yet). Then we apply different proxies to conduct the search with zero-cost PT on the NASBench-201 search space. We report the accuracy of architecture found via different proxies below:
> > |Method|C10|C100|Img16-120|Costs(GPU hours)|
> > |-|-|-|-|-|
> > |Zero-PT+SNIP|93.52±0.18|70.75±0.19|44.45±0.14|0.10|
> > |Zero-PT+NASWOT|93.42±0.07|70.77±0.51|45.11±0.26|0.11|
> > |Zero-PT+Synflow|87.68±0.16|58.92±0.17|32.20±0.00|0.13|
> > |Zero-PT+KNAS|93.95±0.03|72.44±0.26|46.01±0.12|0.10|
> > |Zero-PT+Grad_norm|93.52±0.18|70.75±0.30|44.48±0.11|0.07|
> > |Zero-PT+Grad_norm|93.52±0.18|70.75±0.30|44.48±0.11|0.07|
> > |Zero-PT+Zen-score|93.84±0.05|71.63±0.06|**46.67±0.16**|**0.02**||
> > |Zero-PT+GradSign|93.76±0.12|71.11±0.23|42.95±1.29|0.06|
> > |**Zero-PT+ZiCo(Ours)**|**94.15±0.22**|**72.77±0.66**|46.39±0.23|0.12|
> >
> > As shown above, the architectures found via ZiCo have the highest test accuracy except for Img16-120 datasets (ZiCo is the second best on Img16-120)). More details are given in Appendix F.2 (Page 27).
> >
> >
> > **Q5 Why only show the result of Zen-score partly, ignoring its better results of 83.6% under the 1000M FLOPS constraint?**
> >
> > For Zen-score, the model with 83.6% has 180M parameters and 20G FLOPs, instead of 1G (1000M) FLOPs in our case. Please check the ‘ZenNet-1.2ms’ row in Table 7 on Page 38 of the original [Zen-score-paper](https://arxiv.org/pdf/2102.01063.pdf)).
> > In fact, for the same FLOPs budget of 1000M, our model is similar to Zen-score (i.e., 80.4% vs vs. 80.8%). Moreover, our method performs consistently better than Zen-score on mainstream benchmarks like NASBench201 and NATSBench. Furthermore, ZiCo has up to 1% higher accuracy when training without knowledge distillation (please check the following response).
> >
> >
> > **Q6 How about the results without distillation of Efficient-B3?**
> >
> > We train the networks found via our ZiCo with 150 epochs _without_ any advanced data augmentation techniques (e.g., mixup, random augmentation, or cutout) and _without_ knowledge distillation. We also train Zen-score, and EfficientNet using the exact same setup. The results on shown below:
> >
> > |Method|Top-1/FLOPs|Top-1/FLOPs|Costs(GPU days)
> > |-|-|-|-|
> > |Zen-score| 75.6/410M|76.1/611M|0.5|
> > |EfficientNet|76.0/390M|77.4/700M|3800|
> > |ZiCo|76.5$\pm$0.2/448M|77.1$\pm$0.3/603M|0.4|
> >
> > As shown, without knowledge distillation and advanced data augmentation tricks, ZiCo still outperforms previous SOTA zero-shot NAS Zen-score by up to 1% higher accuracy. Moreover, ZiCo also achieves very similar accuracy to EfficientNets but with 100M fewer FLOPs and 9500x less search time. We provide more comparisons in Table 9 on Page 28.
> >
> >
> > **Q7 Since the proxy ZiCo is defined from the absolute mean values and standard deviation values, it will be interesting to explore their individual contribution.**
> >
> > We provide some empirical analysis to address this question. We randomly select 2000 networks from NATSBench-SSS, run them with CIFAR10, CIFAR100, and Img16-120 datasets and compute the following proxies: (i) Mean value of gradients only; (ii) Standard deviation (STD) value of gradients only; (iii) Combination of mean and STD value, i.e., _ZiCo_. We then calculate the correlation coefficients between these proxies and the real test accuracy:
> >
> > |Proxy|C10(KT/SPR)|C100(KT/SPR)|Img16-120(KT/SPR)|
> > |-|-|-|-|
> > |Mean Only|0.25/0.37|0.39/0.55|0.61/0.81|
> > |STD only|0.39/0.55|0.42/0.6|0.45/0.62|
> > |**ZiCo(Mean+STD)**|**0.54/0.73**|**0.55/0.75**|**0.70/0.88**|
> >
> > As shown above, the contribution of mean or STD individually depends on the dataset. For example, the mean value contributes more than STD on Img16-120, while STD contributes more on CIFAR datasets. In general, by combining mean and STD values together, ZiCo achieves a much higher correlation than using each of them individually. More details are given in Appendix F.1 on Page 27.
> >
> >
> > We hope that our response fully addresses Reviewer’s questions so Reviewer would consider raising the ratings for our work.
> >
> >
> > [1] Xiang, Lichuan, et al. "Zero-Cost Proxies Meet Differentiable Architecture Search." arXiv:2106.06799 (2021).

---

> > > ### Author Response · Authors · 2022-11-16
> > > **Sincerely expecting further discussions with reviewer V9fp**
> > >
> > > Dear Reviewer V9fp:
> > >
> > > We want to thank you for the constructive comments in your review. As a follow-up to our responses, we would like to kindly remind you that the discussion period is ending soon. We hope to use this open response period to discuss the paper to solve the concerns and improve the quality of our paper. Have you gotten a chance to read our responses above, which attempt to address all of your concerns?
> > >
> > > We would be more than happy to provide more information or clarification and hope that they could lead to a positive and fair assessment of this paper.
> > >
> > > Best,
> > >
> > > Authors of paper 2989

---

> > > ### Author Response · Authors · 2022-11-28
> > > **Thanks for increasing the rating**
> > >
> > > Dear reviewer V9fp:
> > >
> > > We deeply appreciate your time and acknowledgment of our previous responses!
> > >
> > > Thanks a lot and best wishes,
> > >
> > > Authors of paper 2989

---

### Author Response · Authors · 2022-11-13
**General Response to All reviewers**

Dear Reviewers, Area Chair, and Senior Area Chair,


We would like to thank you for your constructive feedback on our paper and for giving us a chance to improve our initial submission. The reviewers’ detailed and insightful comments helped us improve this paper tremendously.

A summary of major changes in the revised paper is provided below:

- We provide a new theorem (Appendix C on Page 21) which links our proposed proxy ZiCo to the generalization capacity of neural networks. This way, ZiCo can indicate not only the convergence rate but also the generalization capacity of neural networks. .
- We add five new baseline zero-shot NAS methods mentioned by the reviewers and compare our proxy with them on multiple search spaces and datasets. Detailed side-by-side comparisons are presented in new Table 4 on Page 24.
- We compare our proxy with other methods under two new search spaces: (i) _TransNAS-Bench-101-Mirco_, which consists of more diverse tasks, e.g., image reconstruction and pixel-level prediction  (Appendix E.2 on Page 26), and (ii) _DARTS-based search space_ (Appendix F.4 on Page 28).
- We conduct more extensive ablation studies regarding the training protocols/setups and search algorithms, especially for training without knowledge distillation. (new Tables 7/8/9 on Page 27/28).
- We modified multiple sections of the manuscript following the reviewer's comments and suggestions to improve its readability.

We believe all these changes make this revised manuscript much stronger than the initial submission. All our revisions and new results in the main paper and appendix are highlighted in blue typeface. We hope that our detailed responses to reviewers’ questions address satisfactorily all the issues that have been raised.

We are looking forward to having further discussions with reviewers. Finally, given our extensive updates, we hope that reviewers would consider raising the ratings for our work.

Sincerely,

The Authors of paper 2989

---

### Decision · Program_Chairs · 2023-01-20

**Decision:**

Accept: notable-top-25%

**Justification For Why Not Higher Score:**

This is a nice result but not at the level of an oral presentation

**Justification For Why Not Lower Score:**

The paper proposes one of the best proxies for zero shot NAS

**Metareview: Summary, Strengths And Weaknesses:**


This paper introduces a new zero-cost proxy for neural architecture search (NAS), called ZiCo, motivated by insights on the relationship between gradient statistics, convergence speed and generalization. The proposed proxy is evaluated on NATSBench-TSS, NATSBench-SSS, and NASBench101, and the performance of end-to-end NAS using a MobileNetv2 search space on ImageNet is also evaluated. The authors conduct experiments to validate theoretical claims and conduct ablation studies, such as examining the correlation between ZiCo and accuracy on NATSBench-TSS with varying batch sizes and numbers of batches.

The interaction between the authors and reviewers led to clarifications and improvements to the paper's results and presentation. The additional theoretical analysis added by the authored further motivated the proposed proxy.
I encourage the authors to accommodate the reviewers' comments in the revised version. Moreover, some parts, such as the statement in Section 4 (networks with higher ZiCo tend to have better convergence rates and are therefore better) should be revised in accordance to the reviewers comments.


**Note From Pc:**

if the above contains the word "oral" or "spotlight" please see: "oral" presentation means -> notable-top-5% and "spotlight" means -> notable-top-25%. As stated in our emails, we are disassociating presentation type from AC recommendations